# Unsupervised machine learning reveals key immune cell subsets in COVID-19, rhinovirus infection, and cancer therapy

**Sierra M Barone[1,2†], Alberta GA Paul[3†], Lyndsey M Muehling[3,4†], Joanne A Lannigan[4], William W Kwok[5], Ronald B Turner[6], Judith A Woodfolk[3,4*], Jonathan M Irish[1,2,7*]**

[1]Department of Cell and Developmental Biology, Vanderbilt University, Nashville, United States; [2]Vanderbilt-Ingram Cancer Center, Vanderbilt University Medical Center, Nashville, United States; [3]Allergy Division, Department of Medicine, University of Virginia School of Medicine, Charlottesville, United States; [4]Department of Microbiology, Immunology, and Cancer Biology, University of Virginia School of Medicine, Charlottesville, United States; [5]Benaroya Research Institute at Virginia Mason, Seattle, United States; [6]Department of Pediatrics, University of Virginia School of Medicine, Charlottesville, United States; [7]Department of Pathology, Microbiology and Immunology, Vanderbilt University Medical Center, Nashville, United States

**\*For correspondence:**
jaw4m@virginia.edu (JAW);
jonathan.irish@vanderbilt.edu
(JMI)

[†]These authors contributed
equally to this work

**Competing interest:** See page
19

**Reviewing Editor:** Grégoire
Altan-Bonnet, Memorial Sloan-
Kettering Cancer Center, United
States

**Abstract** For an emerging disease like COVID-19, systems immunology tools may quickly identify and quantitatively characterize cells associated with disease progression or clinical response. With repeated sampling, immune monitoring creates a real-time portrait of the cells reacting to a novel virus before disease-specific knowledge and tools are established. However, single cell analysis tools can struggle to reveal rare cells that are under 0.1% of the population. Here, the machine learning workflow Tracking Responders EXpanding (T-REX) was created to identify changes in both rare and common cells across human immune monitoring settings. T-REX identified cells with highly similar phenotypes that localized to hotspots of significant change during rhinovirus and SARS-CoV-2 infections. Specialized MHCII tetramer reagents that mark rhinovirus-specific CD4+ cells were left out during analysis and then used to test whether T-REX identified biologically significant cells. T-REX identified rhinovirus-specific CD4+ T cells based on phenotypically homogeneous cells expanding by ≥95% following infection. T-REX successfully identified hotspots of virus-specific T cells by comparing infection (day 7) to either pre-infection (day 0) or post-infection (day 28) samples. Plotting the direction and degree of change for each individual donor provided a useful summary view and revealed patterns of immune system behavior across immune monitoring settings. For example, the magnitude and direction of change in some COVID-19 patients was comparable to blast crisis acute myeloid leukemia patients undergoing a complete response to chemotherapy. Other COVID-19 patients instead displayed an immune trajectory like that seen in rhinovirus infection or checkpoint inhibitor therapy for melanoma. The T-REX algorithm thus rapidly identifies and characterizes mechanistically significant cells and places emerging diseases into a systems immunology context for comparison to well-studied immune changes.

## Introduction

Single-cell systems immune monitoring approaches offer new ways to compare how an individual patient's cells respond to treatment or change during infection (*Chattopadhyay et al., 2014*; *Davis*

*et al., 2017*; *Greenplate et al., 2016a*; *Schultze, 2015*). However, the computational analysis tools used widely in this type of systems immunology study were primarily designed to track major cell populations representing >1% of a sample. Viral immune and cancer immunotherapy responses can include mechanistically important and extremely rare T cells that proliferate rapidly over the course of days but as an aggregate exist as <0.1% of blood CD3$^+$ T cells at their peak. These cells can be tracked genetically through clonal expansion, but may be lost in computational analyses focused on describing the global landscape of phenotypes. The specific expansion or contraction of phenotypically distinct cells may be a hallmark feature of key immune effectors and could reveal these cells without the need for prior knowledge of their identity or specialized tracking reagents like MHC tetramers.

The datasets tested here were all suspension flow cytometry, a data type where it is typical to have multiple snapshot samples of cells over time; however, an ongoing challenge in the field is to match or register cells to their phenotypic cognates between samples (*Irish, 2014*; *Pyne et al., 2014*; *Weber and Robinson, 2016*). Analysis algorithms typically rely on aggregate statistics for clustered groups of cells, but the process of grouping the cells works best with larger, established populations (*Diggins et al., 2015*; *Irish et al., 2006*; *Saeys et al., 2016*) and can depend on pre-filtering of cells by human experts (*Greenplate et al., 2016a*; *Greenplate et al., 2019*). Cytometry tools like SPADE (*Bendall et al., 2011*; *Qiu et al., 2011*), FlowSOM (*Van Gassen et al., 2015*), Phenograph (*Levine et al., 2015*), Citrus (*Bruggner et al., 2014*), and RAPID (*Leelatian et al., 2020*) generally work best to characterize cell subsets representing >1% of the sample and are less capable of capturing extremely rare cells or subsets distinguished by only a fraction of measured features. Tools like t-SNE (*Davis et al., 2013*; *Krijthe et al., 2015*), opt-SNE (*Belkina et al., 2019*), and Uniform Manifold Approximation (UMAP) (*Becht et al., 2018*; *McInnes et al., 2018*) embed cells or learn a manifold and represent these transformations as algorithmically generated axes. For a biologist, these tools provide a way to organize cells according to phenotypic relationships that span multiple measured features, such as the proteins quantified on each of millions of cells in the datasets here. In addition to assisting with data visualization, these tools frequently reveal unexpected cells and facilitate their identification through manual or automated clustering (*Davis et al., 2013*; *Becher et al., 2014*; *Diggins et al., 2015*; *Diggins et al., 2017*; *Gandelman et al., 2019*; *Leelatian et al., 2020*). Sconify (*Burns et al., 2018*) is one such tool that applies *k*-nearest neighbors (KNN) to calculate aggregate statistics for the immediate phenotypic neighborhood around a given cell on a t-SNE plot representing data from multiple cytometry samples. This approach to creating a population around every cell was a key inspiration for the Tracking Responders EXpanding (T-REX) tool presented here, which applies KNN to every cell to pinpoint rare cells in phenotypic regions of significant change. In addition to combining UMAP, KNN, and Marker Enrichment Modeling (MEM) in a rapid, unsupervised analysis workflow for paired samples from one individual, T-REX contrasts with prior approaches in its specific focus on the regions of great difference between samples. This T-REX design is based on the observation that, in the absence of a perturbation such as disease or infection, adults tend to have a stable signature of blood cell abundances over weeks to months (*Greenplate et al., 2019*; *Lakshmikanth et al., 2020*; *Mathew et al., 2020*), and the hypothesis that short-term, dramatic changes in rare immune cell subsets will identify cells associated with exposure to an immunogenic agent, such as a virus.

Data types used to challenge the T-REX algorithm here included a new spectral flow cytometry study (Dataset 1) and three existing mass cytometry datasets (Dataset 2, Dataset 3, and Dataset 4). Mass cytometry is an established technique for human immune monitoring where commercial reagents presently allow 44 antibodies to be measured simultaneously per cell (*Greenplate et al., 2016a*; *Mistry et al., 2019*; *Spitzer and Nolan, 2016*). Spectral flow cytometry is gaining attention in human immune monitoring as it generates data that compares well to mass cytometry (*Ferrer-Font et al., 2020*; *Mistry et al., 2019*). Spectral flow cytometers collect cells at around 10-fold the number of cells per second as mass cytometers. While the availability of spectrally distinct antibody-fluorochrome conjugates imposes some practical limits on spectral flow cytometry at present, established panels like the one in Dataset 1 measure ~30 features per cell with excellent resolution, and that capacity is expected to roughly double in the next few years as recent work has demonstrated 40 features (*Park et al., 2020*). Spectral flow cytometry is thus well-matched to studies of very low-frequency cells, as was the case in Dataset 1, where a goal was to computationally pinpoint hundreds of virus-specific T cells in datasets of over 5 million collected cells.

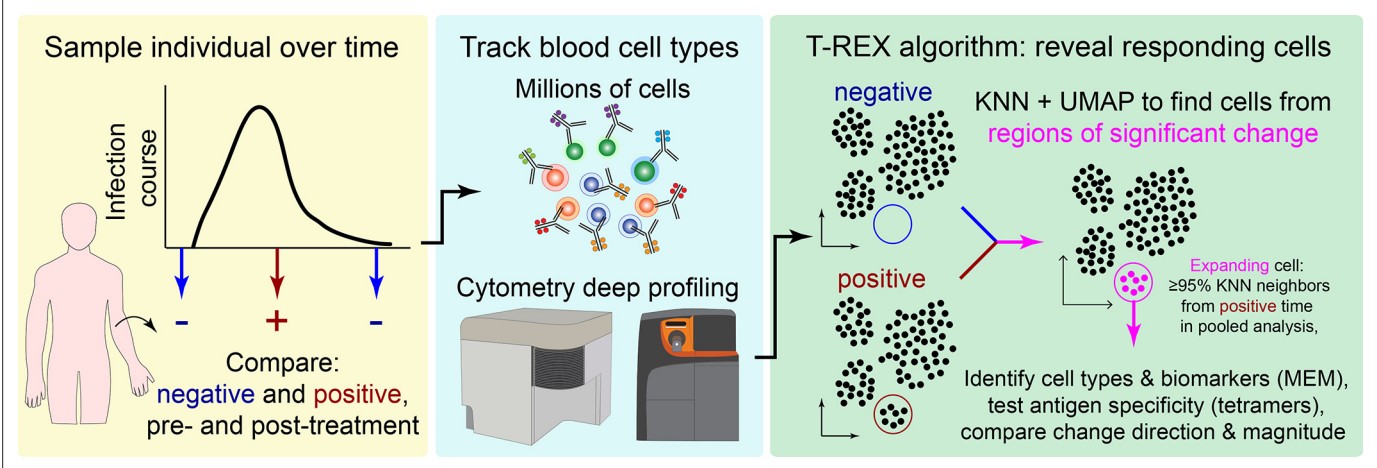

**Figure 1.** Tracking Responders EXpanding (T-REX) algorithm identifies rare cells based on significant expansion or contraction during infection or treatment. Graphic of the T-REX workflow. Data from paired samples of blood from a subject are collected over the course of infection and analyzed by high-dimensional, high-cellularity cytometry approaches (e.g., Aurora or CyTOF instrument, as with datasets here). Cells from the sample pair are then equally subsampled for Uniform Manifold Approximation (UMAP) analysis. A *k*-nearest neighbors (KNN) search is then performed within the UMAP manifold for every cell. For every cell, the percent change between the sample pairs is calculated for the cells within its KNN region. Regions of marked expansion or contraction during infection are then analyzed to identify cell types and key features using Marker Enrichment Modeling. For some datasets, additional information not used in the analysis could be assessed to determine whether identified cells were virus-specific. Finally, the average direction and magnitude of change for cells in the sample was calculated as an overall summary of how the analyzed cells changed between samples.

Datasets 1 and 2 were from individuals infected with two different respiratory viruses, rhinovirus (RV) or SARS-Cov-2, respectively. Respiratory viruses are ubiquitous, and while some, like rhinovirus, are generally benign, they nonetheless pose risks to patients with underlying chronic health conditions. The common colds associated with rhinovirus are characterized by shifts in very rare virus-specific cells in the blood (*Muehling et al., 2020*; *Muehling et al., 2018*). In contrast, novel respiratory viruses, such as SARS-CoV-2, the coronavirus causing COVID-19, continue to emerge that enact high morbidity and mortality, even among healthy subjects. Understanding the immune response to such viruses is vital to treatment and vaccine design, and there has been rapid progress applying human immune monitoring to COVID-19 patients (*Mathew et al., 2020*; *Rodriguez et al., 2020*). An ongoing challenge in the field is to quantitatively compare novel diseases, like COVID-19, to other disease states and immune responses. T cells are pivotal to such responses. Severe COVID-19 has been linked to a pathogenic 'cytokine storm' in which cellular immune responses likely play a crucial role (*Ragab et al., 2020*). Nonetheless, in the case of both rhinovirus and COVID-19, it is clear that host factors are a key determinant of the degree of the T cell response (*Mathew et al., 2020*; *Muehling et al., 2020*). Datasets 3 and 4 were from cancer patients that included melanoma (MB) patients being treated with α-PD-1 checkpoint inhibitor therapies or acute myeloid leukemia (AML) patients undergoing induction chemotherapy. By tracking the CD4$^+$ T cells that expand rapidly during infections and respond to immunotherapy, it may be possible to pinpoint or therapeutically guide cells into helpful vs. harmful roles or niches. Overall, a goal of this study was to develop an automated, quantitative toolkit for immune monitoring that would span a wide range of possible immune changes, identify and phenotype statistically significant cell subsets, and provide an overall vector of change indicating both the direction and magnitude of shifts, either in the immune system as a whole or in a key cell subpopulation.

## Results

We report here T-REX, a novel unsupervised machine learning algorithm for characterizing cells in phenotypic regions of significant change in a pair of samples (*Figure 1*). The primary use case for developing the T-REX algorithm was a new dataset from individuals infected with rhinovirus, where changes in the peripheral immune system are expected in very rare memory cells responding directly to the virus (Dataset 1). Infection with rhinovirus is known to induce expansion of circulating virus-specific

CD4[+] T cells in the blood, and a key feature of the new rhinovirus dataset here is that rare and mechanistically important virus-specific CD4[+] T cells were marked with MHC II tetramers in the context of multiple other T cell markers. The T-REX algorithm was blinded to tetramers during analysis so that they could subsequently be used to test algorithm performance. In addition, T-REX was tested with paired samples from patients with moderate or severe COVID-19 (Dataset 2), melanoma patients being treated with α-PD-1 checkpoint inhibitor therapies (Dataset 3), and acute myeloid leukemia patients undergoing induction chemotherapy (Dataset 4). These datasets were used to determine whether the T-REX algorithm functions effectively across a spectrum of human immune monitoring challenges and to see how the algorithm performs when changes are restricted to rare cell subsets, as in Dataset 1 and Dataset 3, or when many cells may be expanding or contracting, as in Dataset 2 and Dataset 4.

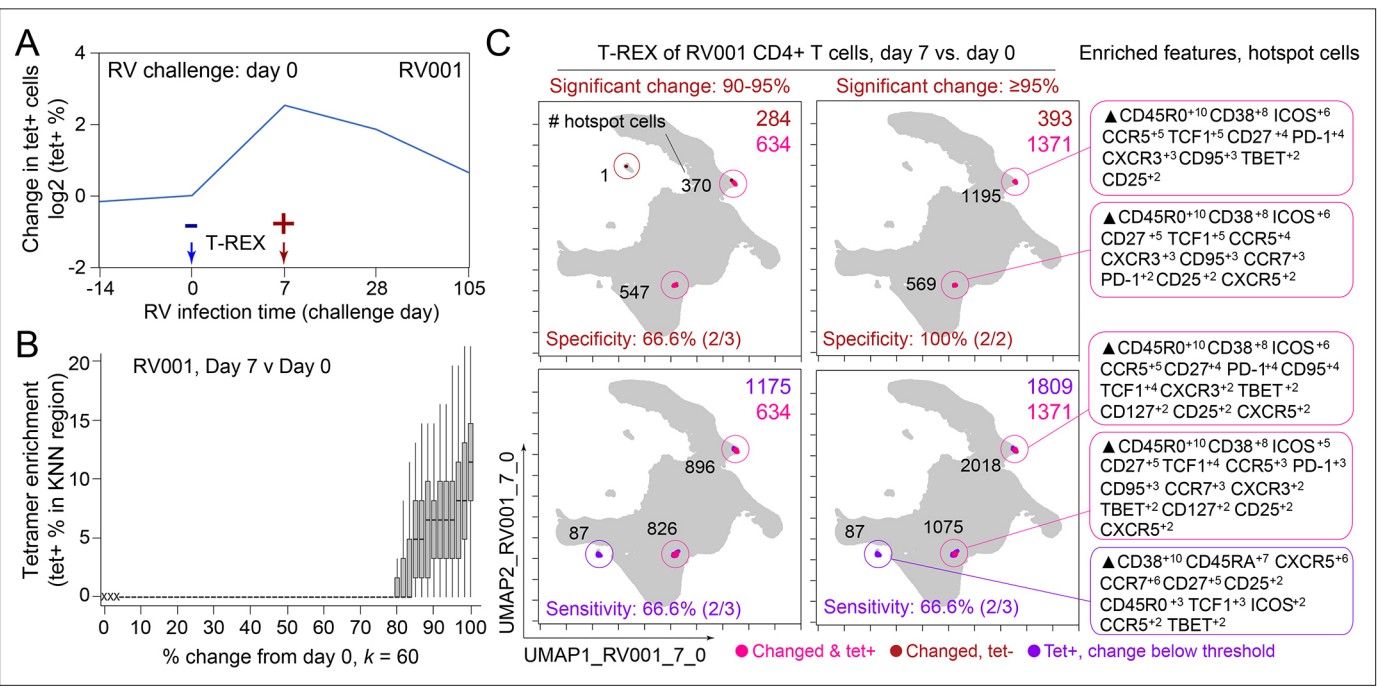

**Figure 2.** Tracking Responders EXpanding (T-REX) identifies molecular signatures of CD4+ T cells that are expanded during acute rhinovirus infection and enriched for virus-specific cells. A subject (RV001) was experimentally infected with rhinovirus (RV-A16) and CD4+ T cell signatures monitored by spectral flow cytometry in conjunction with tetramer staining during the course of infection. (**A**) Fold change in the number of tetramer+ cells (log2) after rhinovirus challenge on day 0. (**B**) Data showing the percentage of tetramer+ cells in each cell's *k*-nearest neighbors (KNN) region (where *k* = 60) plotted against the percentage change in its KNN region on day 7 vs. day 0. A statistical threshold of 80 % or higher for the percentage change in KNN region corresponded to marked enrichment of tetramer+ cells at day 7. (**C**) Uniform Manifold Approximation (UMAP) plots with T-REX analysis of CD4+ T cells for day 7 vs. day 0 based on statistical thresholds of 90–95% change (left column) and ≥95 % change (right column) in cell phenotypes. Pink and red colors denote regions of phenotypic change identified by T-REX. Numbers of tetramer+ cells within the cell's KNN region captured in these areas of phenotypic change are denoted. Cells containing >5% tetramer+ virus-specific cells in the corresponding KNN region are labeled pink. Red cells denote a KNN region that was not enriched for tetramer+ cells, and purple cells denote a tetramer enriched region not captured by T-REX. Values in black indicate the actual number of tetramer+ cells in each circled hotspot of phenotypic change. Marker Enrichment Modeling (MEM) labels on the right indicate cell phenotypes of each hotspot.

The online version of this article includes the following figure supplement(s) for figure 2:

**Figure supplement 1.** Tracking Responders EXpanding (T-REX) identifies CD4+ T cell tetramer+ hotspot using all cells from RV001.

**Figure supplement 2.** Tracking Responders EXpanding (T-REX) identifies regions of great change enriched for tetramers in infected individuals.

**Figure supplement 3.** Marker Enrichment Modeling (MEM)-derived gating strategy for the enrichment of rhinovirus-specific CD4+ T cells.

**Figure supplement 4.** T cell sorting strategy derived using Tracking Responders EXpanding (T-REX) effectively enriches for rhinovirus-specific cells in infected subjects.

# T-REX identifies cells in phenotypically distinct regions of significant change

For the rhinovirus challenge study in Dataset 1, sample pairs available for T-REX included cells taken immediately prior to intranasal inoculation with virus (i.e., pre-infection, day 0), as well as those during (day 7) or following inoculation (day 28). Cells were subsampled equally from each timepoint and then concatenated for a single UMAP specific to each analysis pair. UMAP axes were labeled to indicate they were specific to a comparison for a given individual (*Figure 2*). Thus, each UMAP comparison was a new run of the algorithm. Although it is also possible to map all sample times or all individuals into a single UMAP for analysis, a key goal here was to imagine a minimal T-REX use case with only a pair of samples from one individual. The features selected for UMAP analysis were intentionally limited to surface proteins in order to test whether suitable features for live cell fluorescence-activated cell sorting (FACS) could be identified. Following UMAP, each cell was used as the seed for a KNN search of the local neighborhood within the UMAP axes (i.e., the KNN search was within the learned manifold, as with the analysis in Sconify [*Burns et al., 2018*] or RAPID [*Leelatian et al., 2020*]). The *k*-value for KNN was set to 60 as a starting point based on prior studies and later optimized. For each cell, the KNN region could include cells from either time chosen for analysis, and the percentage of each was calculated to determine the representation of each sampled time in a cellular neighborhood. When cells in regions of expansion (≥95 % of cells in the KNN region from one sampling time) were clustered together in one phenotypic region of the UMAP, they were considered a 'hotspot' of significant change. Cells in change hotspots were aggregated and the phenotype automatically characterized using MEM (*Diggins et al., 2017*). MEM labels here indicated features that were enriched relative to a statistical null control on a scale from 0 (no expression or enrichment) to +10 (greatest enrichment). Ultimately, T-REX and MEM were used to reveal hotspots of ≥95 % change and assign a label that could be used by experts to infer cell identity.

In the human rhinovirus challenge study yielding Dataset 1, MHC class II tetramers were used to identify rhinovirus-specific CD4+ T cells with the goal of tracking phenotypic changes over the course of infection. Increases in tetramer+ cells on day 7 (*Figure 2A*) corresponded to the acute infection phase (*Muehling et al., 2020*). This tetramer tracking system for virus-specific T cells provided an opportunity to test whether the cells identified by T-REX were biologically significant by leaving the tetramer stain features out of the computational analysis (i.e., not using tetramers to make the UMAP or in other parts of T-REX) and then testing to see whether hotspots of cellular change identified by T-REX were statistically enriched for tetramer+, virus-specific cells. A hotspot region was considered to be enriched for tetramer+ cells if >5% of the cells in the region were tetramer+. In the example subject shown, the pairwise comparisons used in T-REX analysis included CD4+ T cells from day 0, immediately prior to rhinovirus infection, and day 7, a well-studied timepoint at which rare, virus-specific CD4+ T cells are observed at higher frequencies (*Muehling et al., 2020*). This trajectory of virus-specific cell expansion was confirmed by a peak in the $\log_2$ fold change in the frequency of tetramer+ CD4+ T cells (*Figure 2A*). Applying T-REX to the rhinovirus data revealed that KNN regions with expansion from day 0 to day 7 were greatly enriched for tetramer+ cells, as compared to regions with less expansion (*Figure 2B*). UMAP axes were labeled as UMAP_RV001_7_0 to denote this UMAP analyzed day 0 and day 7 for individual RV001 (*Figure 2C*). Regions of contraction were observed but were not enriched for tetramer+ cells, except in the case of one individual, RV007, studied here (*Figure 3*). Notably, two of the eight study subjects challenged with rhinovirus were not infected (RV002 and RV003); all other individuals were infected (*Supplementary file 1*). As the focus of this study was virus-specific T cells, CD4+ T cells were the main populations analyzed in all subjects (*Supplementary file 2*). However, T-REX finds the largest tetramer+ cell population with the same phenotype even when using all 4.9 million live cells instead of only the 1.3 million CD4+ T cells (*Figure 2—figure supplement 1*).

A key question for the T-REX algorithm is where to set a statistical cutoff for what is considered to be a biologically significant amount of expansion. Two change cutoffs were tested with subject RV001, ≥90 % and ≥95 % (*Figure 2C*). Using a cutoff of ≥95 % identified 2/2 (100%) tetramer+ hotspots of change for RV001 and did not identify any additional regions that were not tetramer hotspots, whereas the ≥90 % cutoff identified both tetramer+ hotspots and an additional tetramer– hotspot (*Figure 2C*, top). Thus, ≥95 % represented a stringent cutoff that still captured biologically significant cells. An analysis of tetramer enrichment as a function of percentile of expansion from day 0 to day 7

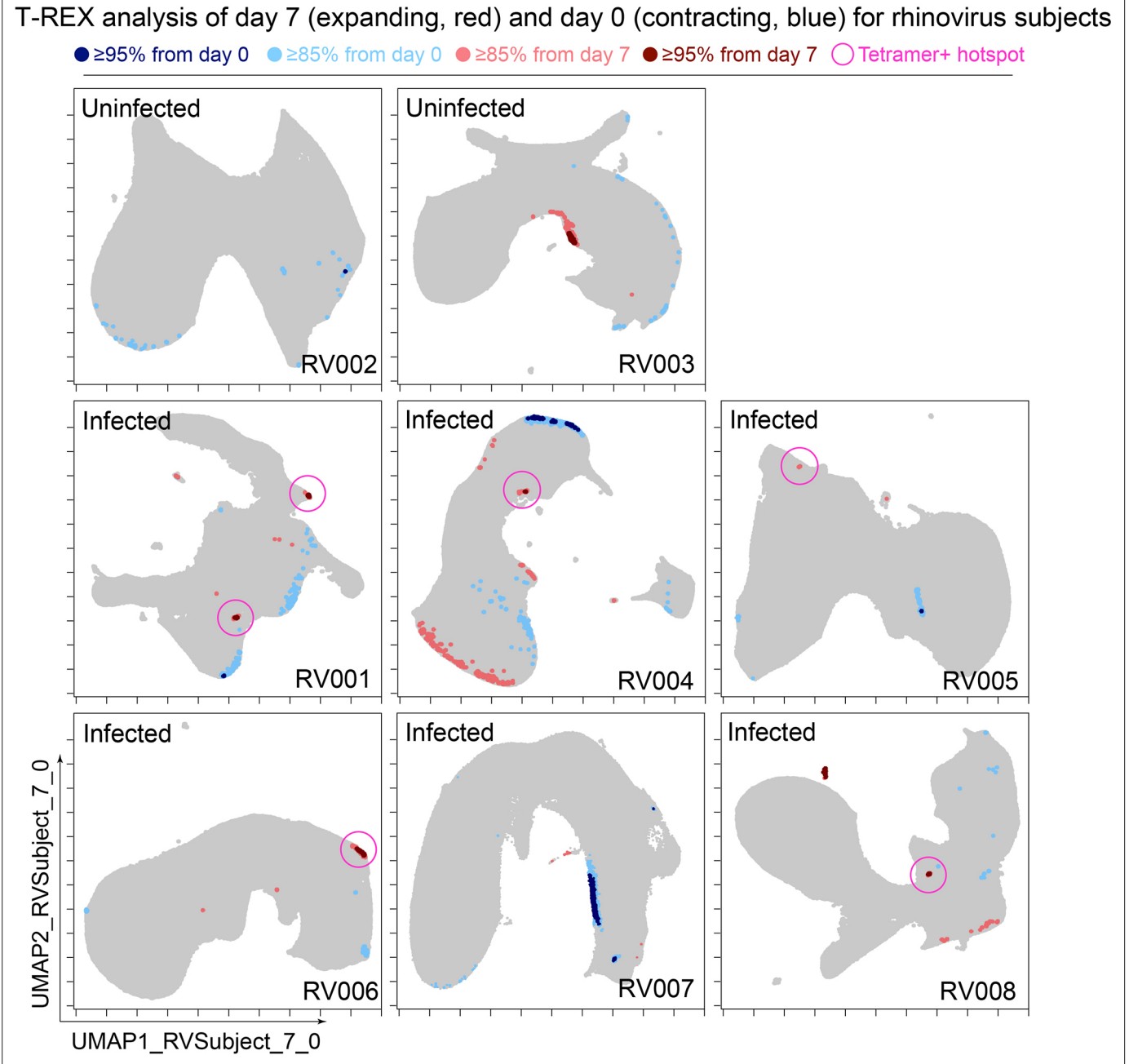

**Figure 3.** Cells in regions of significant change between day 0 and day 7 were typically in tetramer+ hotspots. Tracking Responders EXpanding (T-REX) plots of regions of significant change (blue and red) are shown on Uniform Manifold Approximation (UMAP) axes for CD4+ T cells from eight rhinovirus challenge study individuals. Solid pink circles indicate tetramer+ hotspots that also contained cells that were in regions of marked expansion ≥85%.

(*Figure 2B*) showed that tetramer+ cells were not commonly observed to be in local neighborhoods around cells with change below 80 % in their KNN region. In contrast, above 90 % change, the median CD4+ T cell had 10 % or more tetramer+ neighbors around it in the KNN region (*Figure 2B*). Thus, only regions of 80 % or more expansion from day 0 to day 7 were enriched for tetramer+ CD4+ T cells in study individual RV001.

## A *k*-value of 60 effectively identified immune hotspots in T-REX

A critical question for KNN analysis is the value of *k*, the number of neighbors to assess. While it is useful to have a lower *k*-value as the analysis will complete more quickly, increasing the *k*-value

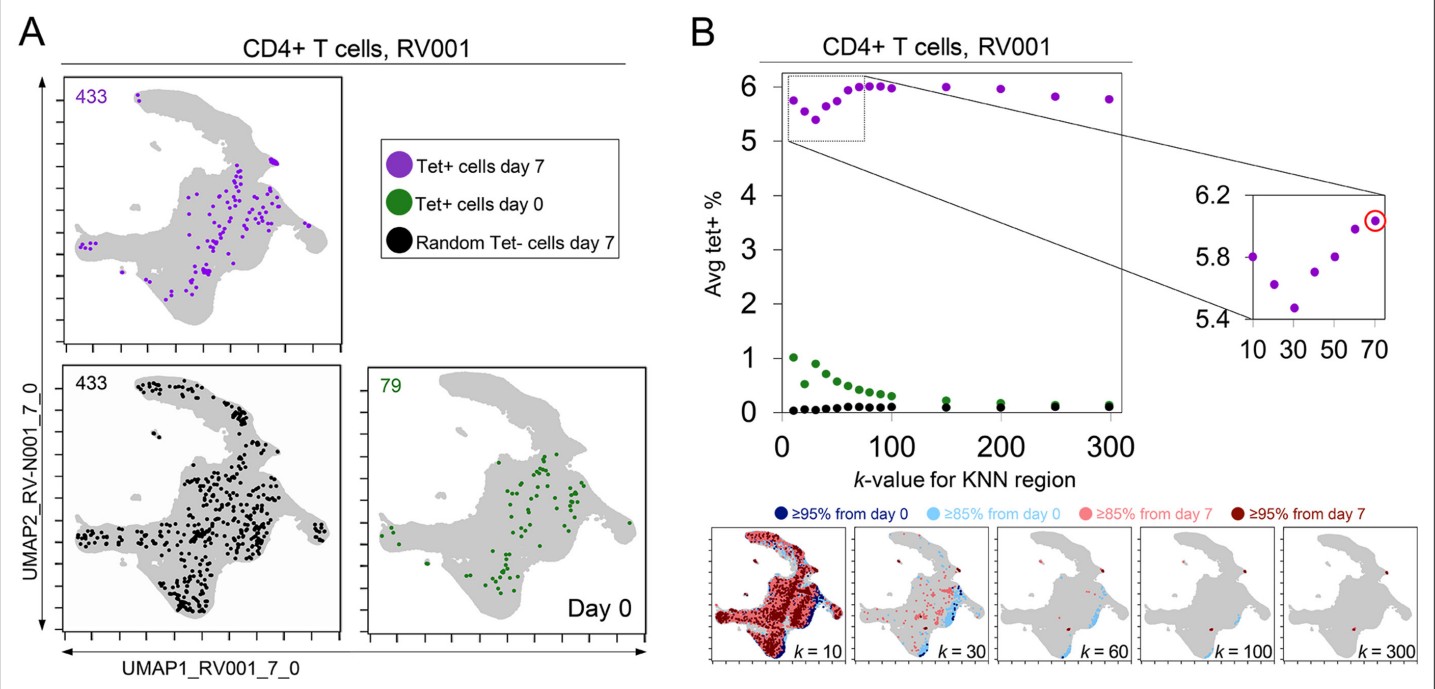

**Figure 4.** k-nearest neighbors (KNN) analysis around tetramer+ cells reveals an optimized k-value at the inflection point of the tetramer density curve. (**A**) Tetramer+ cells from day 7 (dark purple) or from day 0 (light purple) and random tetramer⁻ cells from day 7 (black) are shown overlaid on a common Uniform Manifold Approximation (UMAP) plot. The number of cells for each group is shown in the upper left of each plot. (**B**) Average tetramer enrichment is shown for increasing k-values in repeated KNN analysis of the cells. The inflection point of the resulting curve is circled in red at k = 70, which was the optimized k-value for KNN implemented as in Tracking Responders EXpanding (T-REX) for subject RV001. The T-REX plots on the UMAP axes are shown for various k-values.

might better represent the phenotypic neighborhood or be more statistically robust. To assess how k-value impacted detection of cells in regions of change and the degree to which these cells were virus-specific in rhinovirus challenge Dataset 1, the k-value was systematically changed. In example case RV001, an optimal k was determined to be an inflection point in a graph of the average tetramer enrichment (y-axis, *Figure 4*) vs. increasing values of k (x-axis, *Figure 4*). To calculate this curve, a KNN search was repeated while increasing k in steps from 0 to 300 for every cell in each sampling. This analysis was performed for all tetramer+ cells from day 7 (dark purple, *Figure 4*), all tetramer+ cells from day 0 (light purple, *Figure 4*), and, as a negative control, random tetramer− cells from day 7 (black, *Figure 4*). Within each of these neighborhoods, tetramer enrichment was calculated. This approach identified the inflection point of the percent tetramer+ curve as k = 70 for RV001 (*Figure 4*). In further analysis of the remaining infected rhinovirus subjects, optimal k-values ranged from 30 to 80. A k-value of 60 was chosen and used in all other analyses of rhinovirus subjects (*Figure 3*), as well as Datasets 2, 3, and 4 described below.

## Regions of significant change contained rhinovirus-specific CD4⁺ T cells in Dataset 1

The association between regions of change and enrichment for virus-specific cells observed in the example subject shown (*Figure 2B*) was observed in five infected rhinovirus subjects; tetramer+ CD4⁺ T cells were not enriched in KNN regions around cells that had not expanded from day 0 to day 7 (one infected, two uninfected; *Figure 2—figure supplement 2*). This observation suggested that cutoffs at the 5th and 95th percentile would accurately capture cells representing phenotypic regions with significant change over time. In addition, 15th and 85th percentiles were chosen as cutoffs to capture a more moderate degree of change and track cells that might still be of interest but not from regions experiencing significant change. The remaining cells in phenotypic regions between the 15th and 85th percentiles were not considered to have not changed significantly in the context of these studies. Going forward, it was of interest to determine how often regions of significant change (i.e.,

the 95th and 5th percentile cutoffs) would contain tetramer+ CD4$^+$ T cells in different individuals participating in the rhinovirus challenge study. Cells in regions of significant expansion (≥95%) were also from regions that were enriched for virus-specific cells in nearly all rhinovirus-infected individuals (4/6 at 95 % cutoff, 5/6 at 85 % cutoff) (*Figure 2*, *Figure 2—figure supplement 2*, *Figure 3*). Thus, by focusing specifically on cells in regions representing the most change over time, T-REX analysis revealed subpopulations containing virus-specific cells. This highlights the ability of T-REX to pinpoint such cells without the use of antigen-specific reagents. Following T-REX, MEM analysis was performed using all available features, including intracellular features not used to define the UMAP space (TCF1, TBET, and Ki-67). The phenotype of the regions of significant change enriched for virus-specific cells was quantitatively described with MEM scores (hotspot 1: ▲CD45R0$^{+10}$ CD38$^{+8}$ ICOS$^{+6}$ CCR5$^{+5}$ TCF1$^{+5}$ CD27$^{+4}$ PD-1$^{+4}$ CXCR3$^{+3}$ CD95$^{+3}$ TBET$^{+2}$ CD25$^{+2}$; hotspot 2: ▲CD45R0$^{+10}$ CD38$^{+8}$ ICOS$^{+6}$ CD27$^{+5}$ TCF1$^{+5}$ CCR5$^{+4}$ CXCR3$^{+3}$ CD95$^{+3}$ CCR7$^{+3}$ PD-1$^{+2}$ CD25$^{+2}$ CXCR5$^{+2}$). The change hotspots thus contained activated memory cells (CD45RO + CD38+) that were notable for their early memory/stem-like T cell signature (TCF1+ CD27+), as well as their expression of CCR5 and CXCR3, both of which are chemokine receptors found on rhinovirus-specific CD4+ T cells that respond during infection (*Muehling et al., 2020*; *Muehling et al., 2018*).

To determine the sensitivity of this method, all tetramer+ regions were next reviewed, including those that did not meet the criteria for hotspots of significant change (*Figure 2C*, bottom). In analysis of RV001, 66.6 % (2/3) of tetramer+ regions were captured, meaning there was one region with lower change that contained tetramer+ cells. However, there were only 87 cells in these missed regions compared to 896 cells and 826 cells in the regions with ≥95 % expansion, confirming that T-REX captured the majority of virus-specific cells in the dataset.

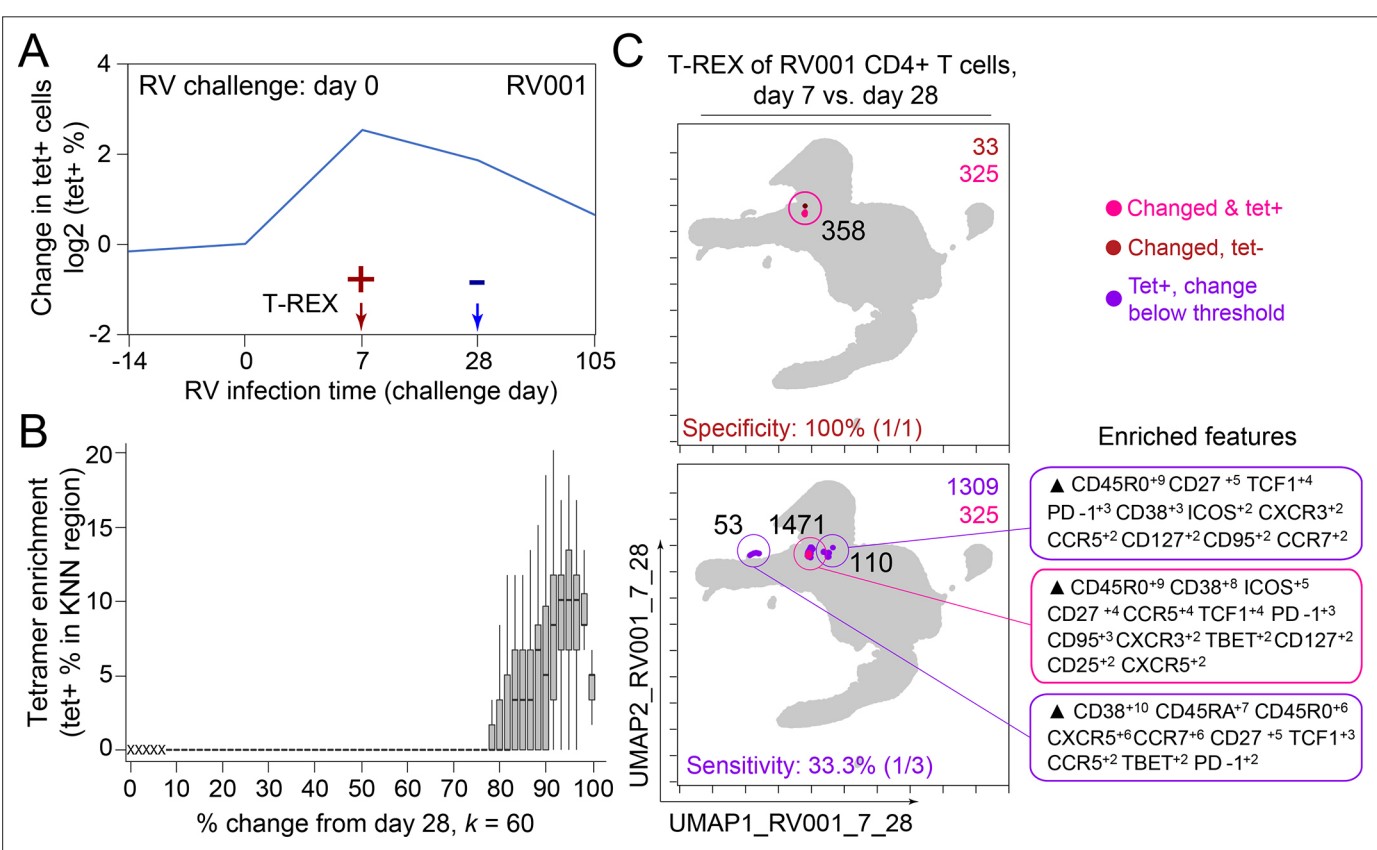

**Figure 5.** Infected cell phenotypes can be compared to cells taken after infection to reveal regions of expansion. (**A**) Fold change in the number of tetramer+ cells (log$_2$) after rhinovirus challenge on day 0. (**B**) Box and whisker plot show *k*-nearest neighbors (KNN) regions in terms of expansion during infection represented by percent change as well as percent of tetramer+ cells for post-infection (day 28) and during infection (day 7). (**C**) Uniform Manifold Approximation (UMAP) plots for 95% change and 5% tetramer cutoffs. Cell count is in black as well as in the upper right of each UMAP plot. Marker Enrichment Modeling (MEM) labels are given for highly expanded and tetramer-enriched regions.

In the case of an emerging infectious disease, it may not be possible to have a pre-infection sample and it would be useful to know whether T-REX analysis of change between a peak of infection and a later time might also reveal virus-specific T cells. To test this idea, pairwise comparisons were performed with cells from day 7 following rhinovirus inoculation and at day 28 after inoculation (*Figure 5*). Strikingly, cells in phenotypic regions of significant change again were enriched for virus-specific, tetramer+ CD4+ T cells. The MEM values for these cells further identified them as CD45R0+ memory cells enriched for CD38, ICOS, CD27, TCF1, CXCR5, PD-1, and CD95 expression, a phenotype matching that of the cells identified in the day 0 to day 7 analysis for this individual (RV001, *Figure 5*).

## Traditional biaxial gating of cells identified by T-REX and MEM enriches for RV-specific T cells

Once identified by machine learning approaches, it can be useful to define a gating scheme that might be used to test whether computationally defined cell subsets can be found using traditional gates. In addition, FACS sorting for live T cells could use surface antigens and biaxial gates to physically separate such cells, as is typical for interrogation in vitro. To test this idea computationally, the features enriched in MEM labels for cells identified by T-REX (MEM label average and standard deviation: ▲CD45R0$^{9\pm2.5}$ CD38$^{7\pm2.0}$ ICOS$^{6\pm1.4}$ CCR5$^{4\pm1.7}$ PD-1$^{4\pm0.9}$ CD95$^{4\pm0.7}$ CD27$^{3\pm1.6}$ CXCR3$^{2\pm0.5}$) were used to define a new gating strategy that used a single positive cutoff gate for each feature of CD4+ T cells (Live, Dump-, CD3+, CD4+) in the order CD45R0, CD38, ICOS, CCR5, PD-1, CD95, CD27, and CXCR3 (*Figure 2—figure supplement 3*). At each gating step, the percentage of RV-specific cells was determined.

It is known that precursor frequencies of rhinovirus-specific CD4+ T cells are very low, even during active infection (0.0004–0.04 % of CD4+ T cells, *Figure 2—figure supplement 4*). The 'virtual sort' successfully enriched for rhinovirus tetramer+ cells in all infected subjects (0.89–9.25 % of CD4+ T cells; *Figure 2—figure supplement 4*). This is notable, considering that the consensus MEM label was generated from regions of ≥95 % change, some of which did not include tetramers. Furthermore, this strategy was able to enrich for tetramer+ cells in the one infected subject for which T-REX was unable to identify tetramer hotspots (RV007), and one in which the tetramer+ hotspot only met a ≥85 % threshold of expansion (RV005), suggesting that T-REX-derived sorting strategies can be broadly applied across cohorts, including subjects whose response may not reach the threshold of identification by T-REX. A minimal panel of 10 markers (Live, Dump−, CD3+, CD4+, CD45R0+, CD38+, ICOS+, CCR5+, PD-1+, CD95+) was sufficient to achieve maximal tetramer enrichment. Interestingly, gating for CD45R0 alone – the first MEM-enriched feature – did not significantly enrich for virus-specific T cells. Furthermore, the T-REX-derived sorting strategy failed to enrich for rhinovirus-specific T cells in uninfected subject, nor did it enrich for CD4+ T cells stained with a control influenza tetramer, confirming the specificity of this method (*Figure 2—figure supplement 4*). Thus, a computational 'virtual sort' for the cells suggests that FACS gates could be drawn using the results of T-REX and MEM analysis. This result further confirms that the populations identified computationally also exist as populations that can be defined in traditional ways.

## T-REX tracking of direction and degree of change contextualizes diverse immune responses

The next goal was to test T-REX with additional data types and contextualize the results from rhinovirus infection (Dataset 1) with changes that might be observed in other immune responses, such as another respiratory infection (Dataset 2), cancer immunotherapy (Datasets 3), or cancer chemotherapy (Dataset 4). To accomplish this, metrics for degree of change as well as direction of change in each sample were devised (*Figure 6A*). Degree of change was calculated as the sum of the percent of cells in the 5th and 95th hotspots of change. Therefore, degree of change is the percentage of the sample that has changed significantly (≥95%) between the two samples (i.e., the percentage of the sample that was dark red or dark blue in the T-REX plot). A degree of change of 100 % would mean that the KNN region around every cell was populated only by cells from the same sample, which is interpreted as meaning the pair of samples had changed completely (half the sample contracting and half expanding). Direction of change is the ratio of ≥95 % expansion to contraction in the sample and was calculated as the difference between the number of cells in the 95th and 5th hotspots of change

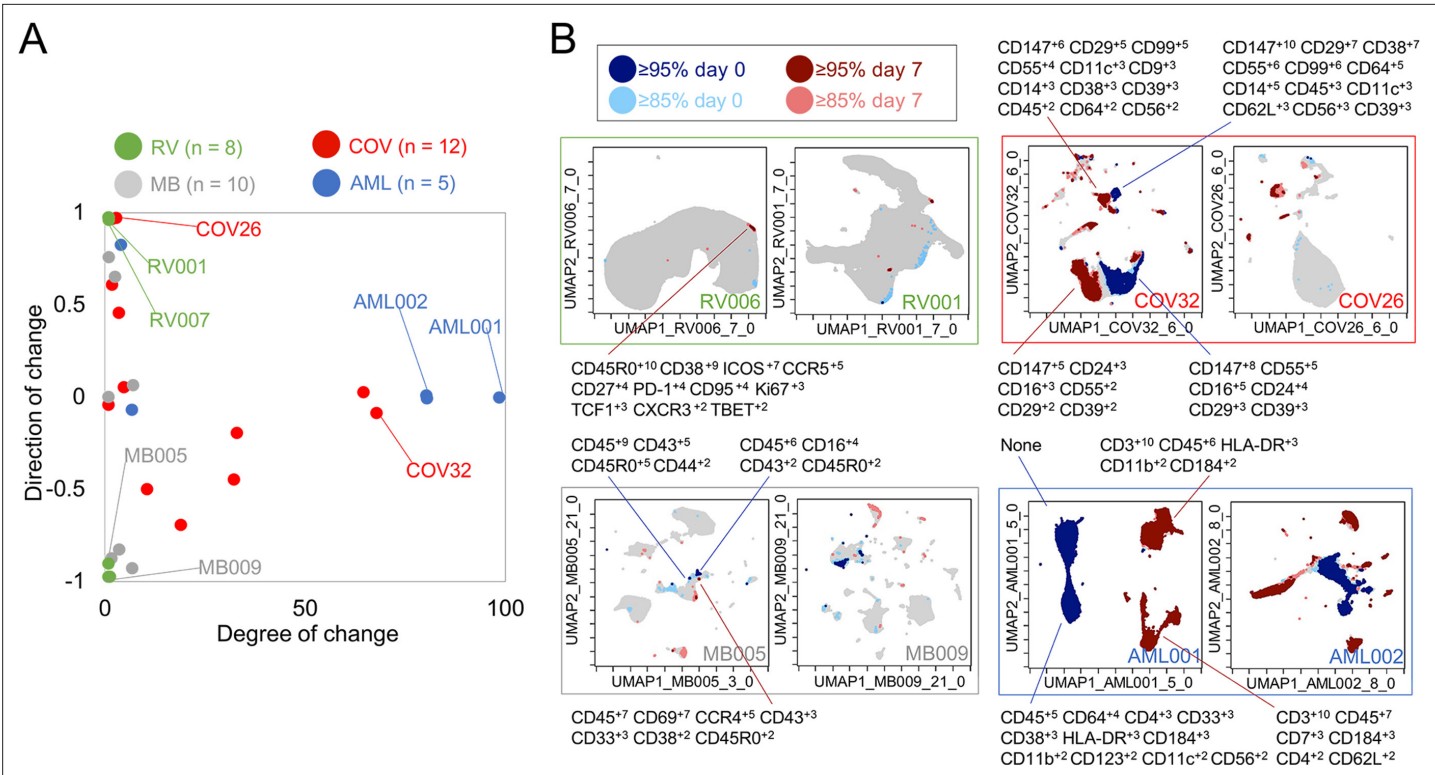

**Figure 6.** Mapping degree and direction of change for 5th and 95th hotspots reveals disease-specific patterns. (**A**) Degree of change and direction of change from Tracking Responders EXpanding (T-REX) analysis in a timepoint comparison shown for acute myeloid leukemia (AML, day 5/8 vs. day 0), COVID (COV, day 1/3/4/5/6/7 vs. day 0), melanoma (MB, day 21/35 vs. day 0), and rhinovirus (RV, day 7 vs. day 0) samples. (**B**) Example T-REX plots are shown for each disease type analyzed. Degree of change shown in red and blue with red showing regions of expansion over time compared to the blue representing regions of contraction over time. Marker Enrichment Modeling (MEM) label given for change hotspots in the left example in each sample type.

The online version of this article includes the following figure supplement(s) for figure 6:

**Figure supplement 1.** Root mean square deviation (RMSD) on Tracking Responders EXpanding (T-REX) hotspot phenotypes from analysis of the COVID-19 CD4+ T cells identified three, distinct phenotypic groups.

**Figure supplement 2.** Marker Enrichment Modeling (MEM)-derived gating strategy for the enrichment of CD4+ T cells in COVID-19-infected individuals.

divided by the sum of the number of cells in the 95th and 5th hotspots of change. A negative direction of change value indicates that there was significant contraction that was more common than significant expansion. The values for directionality range from 0.00 (evenly contracting and expanding) to ±1.00 (all contracting or expanding). This way of looking at the data provided a method for comparing changes across different systems immune monitoring settings (*Figure 6*). Rhinovirus subjects had small changes in samples over time with a median of 0.019 % and an interquartile range (IQR) of 0.0006 %. Rhinovirus also had large directionality across all subjects either up or down, with a median of 0.029 and an IQR of 2.00. Thus, rhinovirus infections resulted in an extremely low magnitude of change, as very rare cell subsets were responding, and the direction of this change was typically fairly high or low for a given individual (i.e., the changes were not balanced and tended to represent marked expansion or contraction in the rare subsets that changed; *Figure 6*).

## Regions of change included cells expressing CD147 and CD38 in COVID-19 Dataset 2

Next, T-REX was applied to Dataset 2, a mass cytometry study of longitudinal collection of blood from patients with COVID-19 (*Rodriguez et al., 2020*). This study originally contained data for 39 total patients, of which 12 patients had accessible mass cytometry data with at least two blood samples over time. For each patient, the day 0 timepoint and the closest sampled timepoint to day 7 were

used for pairwise comparison using T-REX. The COVID-19 samples varied from <1% to 68% in terms of degree of change, with a median of 6.86 % and an IQR of 30.4 %. The directionality of change was near zero, with a median of –0.00880 and IQR of 0.773. Thus, the blood of COVID-19 patients could display marked changes or little change. Notably, the changes <5% were generally positive (median directionality of 0.55, N = 6), whereas the COVID-19 patient cell populations experiencing change >5% typically decreased between day 0 to day 7 (median directionality of –0.33, N = 6).

In T-REX analysis looking at change on the UMAP axes, patients with significant change were apparent due to large islands of cells being painted dark red or dark blue, indicating ≥95 % change between paired days (*Figure 6*). These cell populations were clustered and separated into populations representing day 0 or the later time near day 7, and MEM labels calculated in order to assess the identity and phenotypic changes. For example, patient COV26 saw little change (magnitude of 2.02%) and this was almost entirely expansion (directionality of 0.99). The largest population experiencing significant change from COV26 decreased over time and had a MEM phenotype of $CD147^{+10}$ $CD99^{+8}$ $CD29^{+6}$ $CD38^{+4}$ $CD55^{+3}$ $CD14^{+2}$ $CD39^{+2}$ $CD64^{+1}$ $CD56^{+1}$ $CD8a^{+1}$, indicating that it was a $CD14^+$ myeloid cell subset with high expression of CD147/Basigin. The phenotypes for all automatically identified clusters of cells that expanded or contracted greatly, and the degree and direction of change for each COVID-19 patient from Dataset 2 are listed in *Supplementary file 3*. These reference phenotypes should be comparable to those in other studies of COVID-19, and a meta-analysis of phenotypes could use quantitative analysis of MEM labels to compare these highly expanding and contracting cells.

T-REX was also applied to only $CD4^+$ T cells from patients with sufficient T cell counts (10 out of the 12 patients as described above). Of the 10 COVID-19 patients available for analysis, 5 individuals had at least one hotspot of great change, as revealed by T-REX, in $CD4^+$ T cell-specific analysis (*Figure 6—figure supplement 1*, *Supplementary file 3*). Analysis of the COVID-19 CD4+ T cell hotspot phenotypes using root mean square deviation (RMSD) analysis (*Diggins et al., 2018*; *Diggins et al., 2017*) identified three phenotypic groups. One of these groups was a set of closely related T cell subsets from one individual, patient COV32, and the aggregate MEM label for this population was (MEM label average and standard deviation: $▲CD57^{+8±0.8}$ $CD99^{+9±1.4}$ $CD29^{+7±0.5}$ $CD147^{+6±0.5}$ $CD43^{+5±0.6}$ $CD45^{+4±0.3}$ $CD3^{+4±0.5}$ $CD81^{+4±0.4}$ $CD52^{+4±0.3}$ $CD49d^{+3±0.5}$ $CD45RA^{+3±2.3}$ $CD5^{+3±1}$ $CD56^{+2±1.5}$, *Figure 2—figure supplement 4*). Another phenotype of CD4+ T cells was consistently observed in those COVID-19 patients where T-REX revealed a hotspot. Of the five patients where T-REX identified a CD4+ T cell hotspot, four of the patients had a hotspot matching the aggregate phenotype (MEM label average and standard deviation: $▲CD147^{+9±0.8}$ $CD99^{+9±1.3}$ $CD29^{+8±1.3}$ $CD45^{+4±2}$ $CD3^{+4±0.7}$ $CD38^{+4±1.8}$ $CD49d^{+3±1.6}$ $CD52^{+3±1}$ $CD27^{+3±2.1}$ $CD28^{+3±0.8}$ $CD81^{+2±1.3}$ $CD62L^{+2±1.2}$ $CD56^{+2±0.7}$ $CD5^{+2±0.6}$, *Figure 6—figure supplement 1*). When comparing day 0 to day 6 (±3 days), this population of $CD4^+$ T cells was observed to change significantly in patients COV24, COV29, COV32, and COV39 (four of the five with a $CD4^+$ T cell hotspot). The features of this subset could now be used to physically separate this population using FACS, highlighting a practical application. As a test of this idea, manual biaxial gating, as in standard FACS for physical separation of cell subset, was performed using the cell surface markers identified by MEM as most enriched, as in the prior analysis of cells from the rhinovirus study (*Figure 6—figure supplement 2*). While CD4+ T cells only represented 5.5–14.0% of total cells, following MEM-based gating they were enriched to 49.6–83.3% of cells. Following expert gating, the MEM label of the resulting population was $▲CD147^{+9±0.6}$ $CD99^{+8±1}$ $CD29^{+7±0.6}$ $CD38^{+7±0.8}$ $CD27^{+5±0.8}$ $CD45^{+4±1.2}$ $CD3^{+4±0.6}$ $CD49d^{+3±0.5}$ $CD81^{+3±0.7}$ $CD52^{+3±0.7}$ $CD28^{+3±0.7}$ $CD62L^{+2±0.8}$ $CD56^{+2±0.5}$ $CD5^{+2±0.6}$, which closely matched the computationally identified cells.

## T-REX reveals immune cell changes during cancer therapies in Dataset 3 and Dataset 4

T-REX was next tested on two previously published cancer immune monitoring studies representing a wide range of immune system changes, from modest to extensive. Dataset 3 consisted of mass cytometry analysis of peripheral blood mononuclear cells (PBMCs) from melanoma patients treated with anti-PD-1. This well-studied dataset primarily includes melanoma patients whose blood had modest, subtle shifts in PBMC phenotypes over time. However, one patient in the set, patient MB-009, developed myelodysplastic syndrome (MDS) and experienced a great shift in blood immunophenotype in parallel with the emergence of a small population of blasts in PBMCs (*Greenplate et al., 2016b*).

Overall, when analyzed by T-REX, the melanoma samples in Dataset 3 for comparisons of day 21/35 vs. day 0 had a small degree of change (median of 0.58 % and an IQR 2.34%) with a varying direction-ality (median of –0.42 and an IQR of 1.46), confirming the subtle shifts in phenotypes as previously indicated. The great shift in peripheral immunophenotypes observed in MB-009 was confirmed with T-REX analysis when comparing the 6 -week and 12 -week times. Notably, at 6 weeks, the peripheral blast count was still below 5 % (**Greenplate et al., 2016b**), so T-REX detected a substantial change in subsets that were not driven solely by the emergence from the marrow of the MDS blasts.

Dataset 4 was chosen to represent large changes and included peripheral blood from AML patients treated with induction chemotherapy (**Ferrell et al., 2016**). The compared timepoints for the AML data in Dataset 4 were day 5/8 vs. day 0. As expected, the majority of AML patients had a large degree of change in samples (median of 81.0 % and an IQR of 75.2%) with little to no directionality to the change (median of –0.00250 and an IQR of 0.0173), meaning that there were massive changes in terms of both expansion and contraction over the course of treatment. MEM labels showed that the cells contracting in responder patients were the AML blasts, whereas the emerging cells were the non-malignant immune cells (**Figure 6**). AML samples with a degree of change >80% (AML001, AML002, AML004) came from patients with high blast count in the blood and complete response to treatment, indicating the complete transformation of the immune environment after treatment. AML007, a patient with no blasts in the blood, had a degree of change of 5.97 % over treatment. For AML003, a patient that did not respond to treatment, little change was seen from days 0 to 5 (degree of change = 3.19%) by means of T-REX analysis.

## T-REX outperforms other algorithms based on accuracy and speed

The T-REX workflow combines multiple algorithms, and each step represents a choice of tool and settings. T-REX was next tested against (1) T-REX with different choices of dimensionality reduction or no dimensionality reduction, (2) T-REX with a different clustering method from KNN, and (3) other algorithms. The results were evaluated based on speed (overall runtime, including any dimensionality reduction), whether regions of significant change were detected, and whether regions of change were also tetramer+ hotspots (**Figure 7**).

Throughout this work, we have presented a version of T-REX that utilizes UMAP for dimensionality reduction and KNN for clustering. This standard version of T-REX ran in 0.35 hr on the 1.3 million cell dataset from RV001 (**Figure 7**, left column) and identified a cluster that was 22 % tetramer+. An altered version of T-REX, in which t-SNE replaced UMAP as the method of dimensionality reduction, resulted in a much longer runtime of 23 hr and comparable capture of areas of change, containing 22 % tetramer+ cells (**Figure 7**). Finally, we tested T-REX without dimensionality reduction, running KNN directly on the 14 original surface protein features previously utilized when performing either UMAP or t-SNE dimensionality reduction. This KNN on the original feature space was both slower (1.7 hr) and less accurate, in that it identified too many putative regions of change, most of which were not tetramer+ hotspots (**Figure 7**).

We also compared T-REX with other commonly used dimensionality reduction and clustering algo-rithms (**Weber and Robinson, 2016**), including SPADE, Phenograph, FlowSOM, and Citrus. These algorithms were designed to run on the original measured features and not on t-SNE or UMAP axes, although we have previously used FlowSOM on t-SNE as part of the RAPID algorithm (**Leelatian et al., 2020**). When SPADE was tested using the 14 original features, no regions of change ≥95 % were iden-tified (**Figure 7**). Of the 160 cell clusters requested and produced in SPADE, only two clusters had a percent change between 85% and 90% and contained 7% and 10% tetramer+ cells. The other clusters changed little (average of 53% ± 10.5%). Next, the KNN-based Phenograph algorithm was tested using the 14 original features and a *k*-value of 60 (as in T-REX, **Figure 4**). This test of Phenograph identified 13 clusters, none of which displayed significant change (average of 49% ± 6.4%) and none of which contained >5% tetramer+ cells. Phenograph was also tested using the UMAP coordinates and a *k*-value of 60 (as with standard T-REX in **Figure 2**). This approach identified 113 clusters, one of which was between 85% and 95% change, 8 % tetramer+, and contained cells that were phenotypi-cally similar to those identified by T-REX using MEM (**Figure 7**). The remaining clusters identified by Phenograph did not significantly change (average of 50% ± 6.6% change in the KNN region). Next, FlowSOM was tested both on UMAP axes and original features using an optimized cluster number, as determined by RAPID (**Leelatian et al., 2020**). RAPID cluster optimization (testing 0–90 clusters)

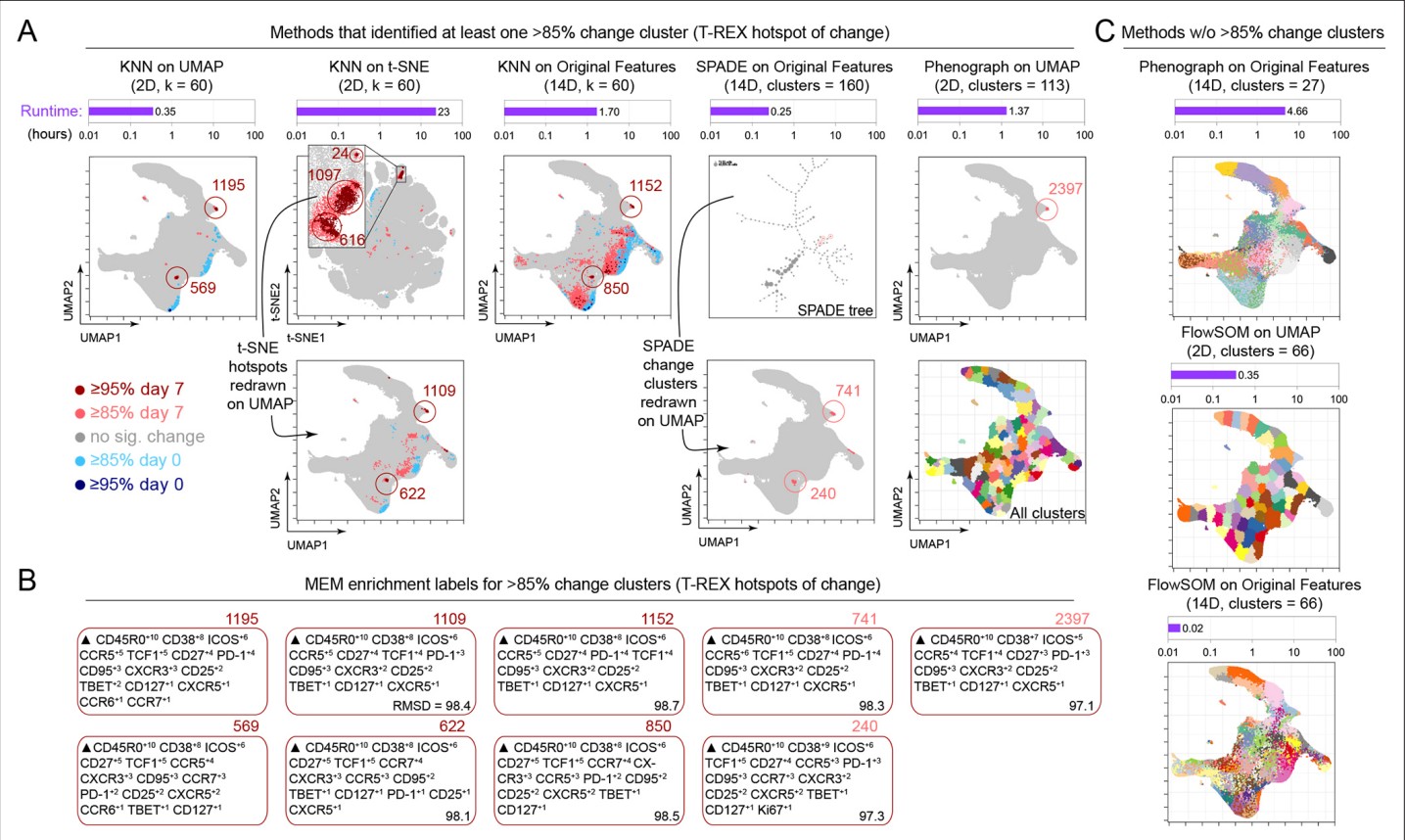

**Figure 7.** Tracking Responders EXpanding (T-REX) using Uniform Manifold Approximation (UMAP) rapidly identified regions of change and tetramer+ hotspots and contrasted with other algorithms. (**A**) Data analysis methods were compared using on RV001 day 0 and day 7 data. Methods identifying at least one cluster with >85% change from day 0 to day 7 are shown. Cells are plotted on UMAP or t-SNE axes (red for expanding cells from day 7, blue for contracting from day 0). Time per method is shown in the purple bar above each UMAP/t-SNE plot. For methods not using UMAP, clusters and hotspots have been redrawn on the UMAP axes to compare to the original T-REX method. Numbers of tetramer+ cells within the cell's *k*-nearest neighbors (KNN) region captured in these areas of phenotypic change are denoted. (**B**) Marker Enrichment Modeling (MEM) labels are shown per method for clusters with significant change. A root mean square deviation (RMSD) score is included in the lower right of each MEM label box for comparison to the original T-REX method MEM labels. (**C**) All clusters on the UMAP axes are shown for methods that did not identify regions of significant change. Runtimes for these are included above each plot in purple.

The online version of this article includes the following figure supplement(s) for figure 7:

**Figure supplement 1.** Tracking Responders EXpanding (T-REX) consistently identified tetramer+ hotspots based on ≥90% or ≥95% change across 10 × 10 % subsamples on a single common Uniform Manifold Approximation (UMAP).

ran in 0.93 hr and identified an optimal value of 66 clusters. FlowSOM did not identify regions of change using either UMAP or original features (average of 50% ± 5%) and did not identify clusters with >5% tetramer+ cells. Finally, we used 10 subsamples of each day to test Citrus on RV001 as Citrus requires two groups and several samples per group. This test of Citrus required 3 hr and found 156 total clusters, with >30 being significantly associated with either day 0 or day 7 abundance. Of these significant clusters, only one had >5% tetramer+ cells. This cluster also closely matched the T-REX tetramer+ cell phenotype (MEM label: ▲ CD45RO$^{+10}$ CD38$^{+8}$ ICOS$^{+7}$ CD25$^{+7}$ Ki67$^{+7}$ CD27$^{+6}$ CCR5$^{+6}$ PD1$^{+6}$ Tcf-1$^{+6}$ CCR6$^{+5}$ CD95$^{+5}$ CCR7$^{+5}$ CXCR3$^{+4}$ CXCR5$^{+4}$ CD127$^{+3}$ Tbet$^{+2}$). Importantly, phenotypes were comparable for all methods that identified at least one hotspot of >85% change, as quantified by RMSD analysis of MEM labels (*Greenplate et al., 2019*) (ranged from 97.1 to 98.7, with 100 being the highest degree of similarity).

Finally, as an additional test of T-REX stability, sample RV001 was subsampled into 10 groups, each containing 10 % of the cells. T-REX was then run using both the same original (common) UMAP (*Figure 7—figure supplement 1*) and subsample-specific (unique) UMAPs. In all cases of subsampling, T-REX consistently identified tetramer+ hotspots based on ≥90 % or ≥ 95 % change across the

10 subsamples. To summarize, T-REX using either UMAP or t-SNE performed the best at identifying regions of change that were greatly enriched for rare tetramer+ cells, and T-REX with UMAP was faster than T-REX with t-SNE (*Figure 7*). Thus, T-REX with UMAP and MEM provides an automated approach to quickly reveal and quantitatively characterize cells in regions of great expansion or contraction.

## Materials and methods

### Experimental rhinovirus infection model

Healthy adult volunteers (ages 18–40) were enrolled in an experimental rhinovirus infection study. All subjects were judged to be seronegative to the challenge virus strain (RV-A16; serum neutralizing antibody titer ≤1:2). Subjects were inoculated with 100 TCID50 of RV-A16 (FDA IND 15162) on study day 0 and were judged to be infected if they seroconverted to the challenge virus by study day 28 (≥4 -fold increase in titer) and/or shed virus in nasal wash specimens during the first  five days of infection, according to standard protocols (*Muehling et al., 2018*). Peripheral blood specimens were obtained sequentially over the course of infection to capture pre-infection immune fluctuations (days –14 and 0), the adaptive phase of infection (day 7), and convalescence (day 28). PBMCs were isolated using density gradient centrifugation and viably cryopreserved for later analysis.

### Rhinovirus tetramer staining and flow cytometry

Rhinovirus tetramer staining was performed as previously described (*Muehling et al., 2020*; *Muehling et al., 2016*; *Muehling et al., 2018*). Briefly, PBMCs were thawed, and all timepoints analyzed together in single experiments. Cells were stained with up to three unique rhinovirus MHCII tetramers, selected to match each individual's HLA type (*Muehling et al., 2016*), and counterstained for viability and surface markers (*Supplementary file 4*). An aliquot of tetramer-labeled cells was enriched using anti-PE magnetic beads for antigen-specific CD4+ T cell frequency calculation, as previously described (*Muehling et al., 2020*; *Muehling et al., 2016*; *Muehling et al., 2018*). Cells were fixed and permeabilized (True Nuclear Fixation and Permeabilization buffers; BioLegend, San Diego, CA, USA), and then stained for intracellular markers. Samples were analyzed using a 3-laser Aurora Northern Lights spectral flow cytometer (Cytek Biosciences, Fremont, CA, USA).

### Generation of Dataset 1

Dataset 1 was a newly generated dataset of PBMCs obtained by longitudinal sampling of healthy volunteers who were challenged intranasally with RV-A16. The study was approved by the University of Virginia Human Investigations Committee, performed in accordance with the Declaration of Helsinki, and registered with ClinicalTrials.gov (NCT02796001). Informed consent was obtained from all study participants. Data were collected and processed at the University of Virginia. Sample collection times were defined by established kinetics of memory effector T helper cell responses (*Muehling et al., 2020*; *Muehling et al., 2018*). Cells were stained with antibodies that target markers of naïve, memory and helper T cells (CCR6, ICOS, CXCR3, CD27, CCR5, TBET, CD45RA, CD45R0, CD95, CXCR5, TCF1, CCR7), and activation and proliferation (CD25, CD38, CD127, Ki-67, PD-1). The marker panel also included up to three MHCII/peptide tetramers to identify virus-specific CD4+ T cells (*Muehling et al., 2016*). Data were collected using a 3-laser Aurora spectral flow cytometry instrument. Additional methodological details can be found on this article's online supplement.

### Data pre-processing

Before testing and evaluating the modular analysis workflow for rare cells, data preprocessing and QC of the data was done on all samples for all timepoints, which included spectral unmixing with autofluorescence subtraction, spill-over correction, and applying scales transformation. An arcsinh transformation was applied to the dataset with each channel having a tailored cofactor based on the instrument used to acquire the data as well as to stabilize variance near zero. Manual gating for clean-up of the data was done by an expert to exclude debris, doublets, and dead cells. As helper T cells were of interest for this RV study, the data analyzed was manually gated for CD3$^+$ CD4$^+$ T cells.

### T-REX algorithm

A modular data analysis workflow including UMAP, KNN, and MEM was developed in R, and scripts for analysis of data in this article are available online (https://github.com/cytolab/T-REX; *Barone and Irish, 2021a*; copy archived at swh:1:rev:5e7ae8d512b36bef58b9e0df2b36a6775e82f734). The dimensionality tool used included UMAP. The default parameter settings for UMAP as found in the uwot package in R were used. Since UMAP analyses were specific to a given individual and pair of samples, UMAP axes were labeled to indicate the individual and comparison being made, as in 'UMAP_RV001_07,' which indicated a comparison of day 0 and day 7 for individual RV001. The KNN search from the fast nearest neighbors (FNN) package was used to find the nearest neighbors for a given cell. For this project, a KNN search was done for every cell using the low-dimensional projection of the data as an input for the neighborhood search. The value for *k*, or the number of nearest neighbors, was determined by an optimization of tetramer enrichment within a neighborhood. Percent change per cell is then calculated based on the abundance of cells from the two samples in the KNN region. For regions expanding or contracting significantly (≤5 % and ≥95 % change), DBSCAN is used to cluster the cells and then MEM is used to quantitatively describe the phenotype.

### MEM analysis of enriched features

Marker Enrichment Modeling from the MEM package (https://github.com/cytolab/mem; *Barone et al., 2021b*; copy archived at swh:1:rev:fc72a290c706c1268678b6300007eb59183af2f9) was used to characterize feature enrichment in KNN region around each cell. MEM normally requires a comparison of a population against a reference control, such as a common reference sample (*Diggins et al., 2017*), all other cells (*Diggins et al., 2018*; *Leelatian et al., 2020*), or induced pluripotent stem cells. Here, a statistical reference point intended as a statistical null hypothesis was used as the MEM reference. For this statistical null MEM reference, the magnitude was zero and the IQR was the median IQR of all features chosen for the MEM analysis. Values were mapped from 0 enrichment to a maximum of +10 relative enrichment. The contribution of IQR was zeroed out for populations with a magnitude of 0.

### A putative FACS gating strategy based on T-REX results

In order to assess the applicability of the T-REX algorithm in the development of follow-up FACS experiments, a sorting strategy was devised based on the results of T-REX and then tested computationally using Datasets 1 and 2 (*Figure 2—figure supplement 3*, *Figure 6—figure supplement 2*). To accomplish this, aggregate MEM scores of T-REX hotspots of ≥95 % expansion were generated for each dataset. Cells were sequentially gated in the order of decreasing MEM feature enrichment, ending with a maximum set of 12 markers, reflecting common capabilities for cell sorting. In Dataset 1, the enrichment of tetramer+ cells was assessed in the populations resulting from putative sort gates, as compared with the total CD4+ T cell population. In Dataset 2, the enrichment of CD4+ T cells was assessed within the total cell population after similar gating using putative sort gates designed algorithmically based on the results of T-REX and MEM.

### Data availability and transparent analysis scripts

Datasets analyzed in this article are available online, including at FlowRepository (*Spidlen et al., 2012*). COVID-19 Dataset 2 (*Rodriguez et al., 2020*) (https://ki.app.box.com/s/sby0jesyu23a65cbgv51vpbzqjdmipr1), melanoma Dataset 3 (*Greenplate et al., 2016b*) (https://flowrepository.org/id/FR-FCM-ZYDG), and AML Dataset 4 (*Ferrell et al., 2016*) (https://flowrepository.org/id/FR-FCM-ZZMC) were described and shared online in the associated manuscripts. Rhinovirus Dataset 1 is a newly generated dataset created at the University of Virginia available on FlowRepository (https://flowrepository.org/id/FR-FCM-Z2VX). Transparent analysis scripts for all four datasets and all presented results are publicly available on the CytoLab GitHub page for T-REX (https://github.com/cytolab/T-REX) and include open-source code and commented Rmarkdown analysis walkthroughs.

## Discussion

A signature feature of the immune system is the ability of rare cells to respond to a stimulus by activating and proliferating, leading to rapid expansion of highly specialized cells that may share

both a distinct phenotype and a clonal origin. Here, we report the design of a new algorithm that can identify and reliably phenotype biologically relevant cell types, including very rare cells, which respond in human disease. The T-REX algorithm was designed to capture phenotypic regions where significant change was occurring between a pair of samples from one individual. The fact that T-REX was able to identify the phenotype of cells whose regions were greatly enriched for virus-specific T cells in rhinovirus Dataset 1 highlighted its ability to pinpoint rare cells responding to infection and closely matches what would be expected based on a current understanding of rhinovirus immunology (*Muehling et al., 2020*; *Muehling et al., 2018*). Specifically, after rhinovirus challenge, expanded regions displayed molecular signatures consistent with activated memory (CD45RO + CD38+) and tissue trafficking (e.g., CCR5 enrichment in the MEM labels) that aligned with our previous findings for rhinovirus-specific CD4+ T cells using manual gating methods and a limited marker panel (*Muehling et al., 2020*; *Wisniewski et al., 2018*). The algorithm also reliably identified memory phenotypes of cells responding to rhinovirus infection, thereby revealing its potential to track the evolution of memory responses in vivo, in addition to defining candidate signatures that might be probed in functional assays. Although comparative phenotyping across time was beyond the scope of this study, it will be of high interest in the future to determine whether the vector of change in specific subsets correlates with additional aspects of disease or complicating host factors, such as allergy and asthma.

The T-REX algorithm also revealed potential new research directions, as there were cells that one might predict would be virus-specific, based on the T-REX enrichment analysis, but which were not enriched for the specific tetramers available here (e.g., *Figure 3*, RV007). Genetic analysis for the clonal origin of cells in such regions might help to determine whether these cells correspond to a clonal response for which a tetramer was not available, or else another type of CD4$^+$ T cell response that may or may not be related to rhinovirus infection, such as a 'bystander' response. Additionally, it will be important to test whether this type of finding holds true for other well-studied viruses for which tetramers are available, such as influenza (*Turner et al., 2020*), and whether these findings extend to MHC class I tetramers and CD8 T cells. It was also striking that in the comparisons of day 7 to either day 0 (*Figures 2 and 3*) or day 28 (*Figure 5*), only the expanding cells (red) were in regions that were also tetramer hotspots. However, despite the focus on expansion in the T-REX acronym, contracting cells will likely also be of biological significance in different disease settings (as with AML) or potentially at different timepoints during the course of an infection, for example, as a result of egress from the circulation in the acute phase, or else transitions in memory and tissue-homing subsets that occur later. This aspect would also be expected to translate to different disease settings such as AML.

Extending the use of T-REX algorithm beyond rhinovirus further highlighted its ability to identify responding cells in a consistent manner across different subjects and different disease settings. Indeed, it is notable that regardless of the disease context the patient served as an effective baseline for comparison and allowed T-REX to find phenotypically similar cells in individuals with different starting immune profiles. A central question in systems immune monitoring is to place newly emerging diseases into the context of other well-studied diseases and immune responses. In working to compare COVID-19 and rhinovirus, it became clear that a summary of change indicating both the direction and magnitude of shifts was needed (*Figure 6*). This framework represents a way to summarize both broad populations of immune cells, like all CD45$^+$ leukocytes, and key cell subpopulations, like CD4$^+$ T cells. The striking changes observed in patients with moderate and severe COVID-19 were far beyond the subtle changes observed in individuals with rhinovirus and more closely matched the immune reprogramming observed in melanoma patients receiving checkpoint inhibitor therapy (*Figure 6*). A primary finding of T-REX analysis of Dataset 2 from the blood of COVID-19 patients was that some patients experienced very large changes in the blood, and that these changes were typically associated with more decreases than increases (*Figure 6*). This finding closely matches reported findings from others who observed a systematic reprogramming in many immune cell populations in severe COVID-19 patients (*Mathew et al., 2020*). Also observed were T cell subsets with enrichment of CD38, PD-1, and CD95, as has also been reported. While disease severity is not available for individual patients from Dataset 2, it is known that all these cases were at least moderate or severe (*Rodriguez et al., 2020*). It will be of interest to test the hypothesis that the more severe cases will be one of the two groups, either the patients with very little change and just expansion of cells, or those with more marked change and a general decrease of cells (*Figure 6*).

Notably, CD147/Basigin was highly expressed on many cells that changed during infection and was observed to change greatly on some populations over time. CD147 has been proposed in pre-prints as both a binding partner for SARS-CoV-2 spike protein and a potential mechanism of cellular entry, although evidence is needed to support this controversial hypothesis (*Shilts and Wright, 2020*). In the study of Dataset 2, the authors noted that immune responses were dominated by cells expressing CD38 and CD147 (*Rodriguez et al., 2020*). In the T-REX analysis of the same Dataset 2, for the cells that were changing greatly, CD147 was sometimes present on cells from day 0 that decreased greatly and was lower or absent on cells that emerged only at later times (*Figure 6*). An example of this was seen in cells from patient COV40, for which the authors noted CD147 expression on effector subsets at 1 week and onwards. The cells pinpointed by T-REX as emerging at day 6 included B cells that expressed CD147 (e.g., $CX3CR1^{+8}$ $CD9^{+8}$ $CD29^{+8}$ $\underline{CD147^{\pm5}}$ $IgD^{+3}$ $CD99^{+3}$ $CD33^{+1}$ $CD11c^{+1}$ $HLA\text{-}DR^{+1}$ $CD24^{+1}$, *Supplementary file 3*), but the level of enrichment was lower than on myeloid cells from day 0 that decreased over time (e.g., $\underline{CD147^{\pm8}}$ $CD29^{+6}$ $CD55^{+5}$ $CD38^{+5}$ $CD99^{+4}$ $CD64^{+3}$ $CD62L^{+2}$ $CD45^{+1}$ $CD33^{+1}$ $CD14^{+1}$, *Supplementary file 3*). This pattern of decreased enrichment of CD147 on cells emerging after day 0 was seen on other patients (*Supplementary file 3*) and is consistent with multiple explanations. Overall, there was a strong downward trend in many of the markers and cell subsets in COVID-19 patients, suggesting either selection against cells expressing a high level of proteins, downregulation of expression of key surface markers like CD147, expansion of immature or abnormal cells, or extreme trafficking of cells into tissues. These potential outcomes cannot be distinguished from each other with the analysis here. The utility of the T-REX algorithm is primarily in generating these hypotheses automatically and in pinpointing cells with extreme behavior within the context of the patient as their own baseline. Given the large amounts of change (*Figure 6*) and the generally lower numbers of T cell subsets observed in COVID-19 than in healthy individuals (*Supplementary file 3*), it may be the case that therapeutic stabilization of the immune system will be needed before virus-specific T cells will be identifiable with the T-REX method. It will be especially interesting to explore more mild cases of COVID-19 with this approach and determine whether the hotspots of change are truly virus-specific, analogous to the scenario with rhinovirus.

For the melanoma and AML cases presented here, the cohort sizes were not large enough to allow robust statistical comparison of patient response to degree or direction of change, although this information is available in the original studies (*Ferrell et al., 2016*). Of the AML patients, those with a high magnitude of change (*Figure 6*) were also those that had a high blast count and were complete responders to induction therapy, suggesting that the change represents the overall 'reset' of the immune system following chemotherapy. It will be of high interest to ask whether the identification of virus-specific T cells extends to populations of cells on checkpoint inhibitor therapy. The dynamics of regulatory cells may also be of interest, especially for autoimmunity, and it is possible, but not known, whether these cells will follow the same pattern as the $CD4^+$ T cells in rhinovirus infection.

A major strength of the algorithm is that once cell regions of change are identified, the key features highlighted by T-REX and MEM can be used in lower parameter flow cytometry or imaging panels to provide further information, confirm findings, and physically isolate cells by FACS (*Figure 2—figure supplement 3*). Thus, low parameter cytometry approaches may rely more on manual analysis methods and cell signatures that are determined a priori, and T-REX may provide a useful tool for narrowing in on such features using exploratory high-dimensional data. The computational approach here emphasizes unsupervised UMAP and KNN clustering and uses statistical cutoffs to guide the analysis. Further optimization of the algorithm could include a stability testing analysis where the stochastic components of the algorithm are repeated to determine whether clusters or phenotypes are stable (*Leelatian et al., 2020*; *Melchiotti et al., 2017*). Overall, the unsupervised approach aims to diminish investigator bias and reveal novel or unexpected cell types. While unsupervised analysis tools have impacted high-dimensional cytometry for at least a decade (*Davis et al., 2013*; *Becher et al., 2014*; *Bendall et al., 2011*; *Diggins et al., 2015*; *Saeys et al., 2016*), T-REX is designed to capture both very rare and very common cell types and place them into a common context of immune change. The extremely rare T cells identified here were overlooked by other tools, likely due to these tools typically seeking clusters of cells representing at least 1% and generally more than 5% of the sample. This observation was expected and is consistent with past tests of algorithms on detection of rare subsets, which found that over half the algorithms tested returned poor F1 accuracy scores for detecting known populations that were 0.8% or 0.03% of the cells in a sample (*Weber and Robinson,*

*2016*). Notably, FlowSOM was a better performing algorithm in those examples. Here, FlowSOM did not effectively detect the rare 0.02% of virus specific cells in analyses of original features or dimensionality reduction results (*Figure 7*). An aspect of data structure in this study is the phenotypic similarity within the CD4+ T cell population, which means that the 0.02% of target cells is connected to a larger population and thus less apparent in the raw data. Thus, it is the threshold of change in T-REX removing the vast majority of cells (i.e., the cells that did not expand or contract ≥95%) that makes the virus-specific cells appear as separate clusters and greatly simplifies the task of distinguishing cluster groups.

T-REX was designed for use on a pair of data files, such as the pairs of timepoints tested here, but it can also be used for other paired comparisons. In the rhinovirus examples here, T-REX revealed virus-specific cells based on change over time only. However, the dataset timepoints in the rhinovirus study were selected carefully using prior knowledge to best reveal antigen-specific T cells (i.e., it was known that 7 days would be a good timepoint). In other disease settings, one may not know ideal timepoints and so it is not necessarily the case that T-REX will always reveal antigen-specific cells. It may also be the case that the level of change is so great that it obscures mechanistically significant rare cell subsets. This is an interpretation of the COVID-19 infection examples presented here, wherein a high degree of overall change was apparent in the immune system and many cells are changing that are not antigen-specific T cell subsets. This could potentially be due to timepoint selection or the natural history of the disease. However, we would hypothesize that COVID-19 vaccine responses might be more specific and localized to key cell subsets and, if this is the case, that T-REX would pinpoint key cell populations responding to the vaccine. For studies involving multiple timepoint tests, when it is possible to run a large common UMAP on a pool of the samples (i.e., there are not batch effects), T-REX can be used to do systematic paired comparisons of every individual against the pool. Data from different panels should not be used as the paired samples must have all channels in common. T-REX is also very sensitive to batch effects, and issues like these need to be addressed before analysis, or samples need to be analyzed in a batch-specific manner. We expect that one advantage of using MEM labels to summarize the phenotype of T-REX populations is that MEM labels have been shown to allow quantitative comparison across batches, instruments, and data types (*Diggins et al., 2017*). T-REX performance may depend on the number of cells and the algorithm was designed for studies with hundreds of thousands to millions of cells. Analyses with fewer than 1000 cells are unlikely to be productive, and analyses with fewer cells may require testing and adjusting $k$ more than analyses with >100,000 cells. In testing subsamples, tetramer+ hotspots were still detected with 100,000 cells per sample (*Figure 7—figure supplement 1*). While equal subsampling is part of the algorithm of T-REX, subsampling tested here did not significantly impact the results. So while sampling and rarity of cells may impact the ability of T-REX to find subsets, the method was stable with far fewer cells (10%) than were originally planned as needed in the original study (*Figure 7—figure supplement 1*). The parameters that are chosen by the user are the $k$-value, the markers used to create the UMAP, the markers used for MEM, and the DBSCAN settings. Additional optimization of the $k$-value and cutoffs based on a formal statistical test is something that should be explored in further study of T-REX and related algorithms.

It is a central goal of systems immunology to map people with vastly contrasting immune system changes onto a common plot of change (as in *Figure 6*). The approach here goes beyond prior single measures of systematic change, such as Earth Mover's Distance (*Greenplate et al., 2019*; *Orlova et al., 2016*), by including both direction and magnitude of change in one view of an individual's immune response. This improvement proved useful for comparing settings with great change in many cell types (COVID-19 infection, AML chemotherapy responders) to settings with rare cells that specifically expanded or contracted (rhinovirus infection, melanoma checkpoint inhibitor therapy). This sensitivity of T-REX for extremely rare cells allowed the algorithm to reveal virus-specific CD4+ T cells without prior knowledge of their phenotype. T-REX should now be tested further to determine whether cells identified in SARS-CoV-2 also share a clonal origin. In addition, it is likely that T-REX will be useful beyond immunology settings in paired comparisons of quantitative single cell data, such as discovery screening or paired analysis of tumor cells.

## Acknowledgements

We thank Monika Grabowska at Vanderbilt University for helpful work during a rotation on code that became part of the T-REX algorithm. We also thank Dr. Todd Bartkowiak for useful help and expertise on implementing various additional computational tools for T-REX comparisons. Research was supported by the following funding resources: U01 AI125056 (SMB, AGAP, LMM, RBT, JAW, and JMI), U01 TR002625 (JMI, SMB), R01 CA226833 (JMI, SMB), R21 AI138077 (JAW, LMM), U54 CA217450 (JMI), T32 AI007496 (LMM), the Michael David Greene Brain Cancer Fund (JMI), and the Vanderbilt-Ingram Cancer Center (VICC, P30 CA68485).

## Additional information

### Competing interests

Alberta GA Paul: became an employee of Cytek Biosciences, Inc after performing this research at University of Virginia.. Joanne A Lannigan: became a paid consultant of Cytek Biosciences, Inc after performing this research at University of Virginia.. Jonathan M Irish: was a co-founder and a board member of Cytobank Inc and received unrelated research support from Incyte Corp, Janssen, and Pharmacyclics.. The other authors declare that no competing interests exist.

### Funding

| Funder | Grant reference number | Author |
|---|---|---|
| National Institutes of Health | U01 AI125056 | Sierra M Barone<br>Alberta GA Paul<br>Lyndsey M Muehling<br>Ronald B Turner<br>Judith A Woodfolk<br>Jonathan M Irish |
| National Institutes of Health | R01 CA226833 | Sierra M Barone<br>Jonathan M Irish |
| National Institutes of Health | U54 CA217450 | Jonathan M Irish |
| National Institutes of Health | T32 AI007496 | Lyndsey M Muehling |
| Vanderbilt-Ingram Cancer Center | P30 CA68485 | Sierra M Barone<br>Jonathan M Irish |
| National Institutes of Health | R21 AI138077 | Lyndsey M Muehling<br>Judith A Woodfolk |
| National Institutes of Health | U01 TR002625 | Sierra M Barone<br>Jonathan M Irish |
| Vanderbilt University Medical Center | Michael David Greene Brain Cancer Fund | Sierra M Barone<br>Jonathan M Irish |

The funders had no role in study design, data collection and interpretation, or the decision to submit the work for publication.

### Author contributions

Sierra M Barone, Conceptualization, Formal analysis, Software, Validation, Visualization, Writing – original draft, Writing – review and editing; Alberta GA Paul, Joanne A Lannigan, Data curation, Methodology, Writing – review and editing; Lyndsey M Muehling, Data curation, Formal analysis, Visualization, Writing – review and editing; William W Kwok, Ronald B Turner, Methodology, Writing – review and editing; Judith A Woodfolk, Data curation, Investigation, Methodology, Project administration, Resources, Supervision, Writing – review and editing; Jonathan M Irish, Conceptualization, Formal analysis, Methodology, Project administration, Resources, Software, Supervision, Validation, Visualization, Writing – original draft, Writing – review and editing

### Author ORCIDs

Sierra M Barone http://orcid.org/0000-0001-5944-750X

Alberta GA Paul (iD) http://orcid.org/0000-0002-9318-3760
Lyndsey M Muehling (iD) http://orcid.org/0000-0003-3203-3264
Joanne A Lannigan (iD) http://orcid.org/0000-0002-3981-8681
William W Kwok (iD) http://orcid.org/0000-0003-4843-4599
Judith A Woodfolk (iD) http://orcid.org/0000-0002-8915-4334
Jonathan M Irish (iD) http://orcid.org/0000-0001-9428-8866

### Ethics

Clinical trial registration NCT02796001.

Human subjects: Dataset 1 was a newly generated dataset of PBMCs obtained by longitudinal sampling of healthy volunteers who were challenged intranasally with RV-A16. The study was approved by the University of Virginia Human Investigations Committee, performed in accordance with the Declaration of Helsinki, and registered with ClinicalTrials.gov (NCT02796001). Informed consent was obtained from all study participants. Data were collected and processed at the University of Virginia.

### Decision letter and Author response

Decision letter https://doi.org/10.7554/eLife.64653.sa1
Author response https://doi.org/10.7554/eLife.64653.sa2

## Additional files

### Supplementary files

• Supplementary file 1. Direction and degree of change for all samples as in Barone et al. *Figure 6*. Day compared indicates the actual day a sample was taken relative to a day 0 comparison in all cases. # cells indicates the number of cells in each percentile grouping (e.g., [0,5] indicates the number of cells whose *k*-nearest neighbors [KNN] area had ≤5 % change in Tracking Responders EXpanding [T-REX] analysis). Change magnitude and direction are as in *Figure 6A*.

• Supplementary file 2. Tetramer+ cell frequency in CD4+ T cells in RV subjects. CD4+ T cell and tetramer+ cell counts per subject per day in the RV study. Frequency given as a percentage.

• Supplementary file 3. Marker Enrichment Modeling (MEM) labels for enriched features in all cell hotspots and in CD4+ T cell hotspots from COVID-19 patients in Dataset 2. Red MEM labels denote that the hotspot is a region with ≥95 % expansion, and blue MEM labels denote that the hotspot is a region with ≥95 % contraction after Tracking Responders EXpanding (T-REX) analysis when comparing day 0 to day 4 (±3 days). Labels given for all cell hotspots and CD4+ T cell hotspots.

• Supplementary file 4. Fluorescent antibody panel and tetramer selection for the analysis of RV-specific CD4+ T cells.

• Transparent reporting form

### Data availability

Datasets analyzed in this manuscript are available online, including at FlowRepository. COVID-19 Dataset 2 (https://ki.app.box.com/s/sby0jesyu23a65cbgv51vpbzqjdmipr1), melanoma Dataset 3 (https://flowrepository.org/id/FR-FCM-ZYDG), and AML Dataset 4 (https://flowrepository.org/id/FR-FCM-ZZMC) were described and shared online in the associated manuscripts. Rhinovirus Dataset 1 is a newly generated dataset created at the University of Virginia available on FlowRepository (FR-FCM-Z2VX available at: https://flowrepository.org/id/FR-FCM-Z2VX). Transparent analysis scripts for all four datasets and all presented results are publicly available on the CytoLab Github page for T-REX [(https://github.com/cytolab/T-REX) copy archived at https://archive.softwareheritage.org/swh:1:rev:5e7ae8d512b36bef58b9e0df2b36a6775e82f734] and include open source code and commented Rmarkdown analysis walkthroughs.

The following dataset was generated:

| Author(s) | Year | Dataset title | Dataset URL | Database and Identifier |
|---|---|---|---|---|
| Barone SM | 2020 | Rhinovirus data from: Unsupervised machine learning reveals key immune cell subsets in COVID-19, rhinovirus infection, and cancer therapy | https://flowrepository. org/id/FR-FCM-Z2VX | FlowRepository, FR-FCM-Z2VX |

The following previously published datasets were used:

| Author(s) | Year | Dataset title | Dataset URL | Database and Identifier |
|---|---|---|---|---|
| Rodriguez L | 2020 | COVID-19 data from: Systems-Level Immunomonitoring from Acute to Recovery Phase of Severe COVID-19 | https://ki.app.box. com/s/sby0jesy u23a65cbgv51vpbz qjdmipr1 | COVID-19_project Brodin lab accession, sby0jesyu23a65cbgv51vpbzqjdmipr1 |
| Greenplate AR | 2016 | Melanoma data from: Mass Cytometry of Peripheral Blood from Melanoma Patients Receiving anti-PD-1 | http:// flowrepository.org/ id/FR-FCM-ZYDG | FlowRepository, FR-FCM-ZYDG |
| Ferrell PB | 2016 | AML data from: High-Dimensional Analysis of Acute Myeloid Leukemia Reveals Phenotypic 684 Changes in Persistent Cells during Induction Therapy | http:// flowrepository.org/ id/FR-FCM-ZZMC | FlowRepository, FR-FCM-ZZMC |

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
