## [Decision Letter]

**Acceptance summary:**

The T-REX method characterizes rare populations by cytometry. The potential applications in the context of analyzing antigen-specific T cells (as identified as tetramer-positive cells) are important with an interesting use of 2 timepoints-cohort of samples from rhinovirus-infected patients.

**Decision letter after peer review:**

Thank you for submitting your article "Unsupervised machine learning reveals key immune cell subsets in COVID-19, rhinovirus infection, and cancer therapy" for consideration by *eLife*. Your article has been reviewed by 3 peer reviewers, and the evaluation has been overseen by a Reviewing Editor and Aleksandra Walczak as the Senior Editor. The following individuals involved in review of your submission have agreed to reveal their identity: Stephen De Rosa (Reviewer #2); Ahmed Mahfouz (Reviewer #3).

*Reviewer #1 (Recommendations for the authors):*

The machine learning workflow Tracking Responders EXpanding (T-REX) was created to identify changes in both very rare and common cells in diverse human immune monitoring settings. T-REX analysis of paired blood samples provides an approach to rapidly identify and characterize mechanistically significant cells and to place emerging diseases into a systems immunology context. Data types used to challenge the T-REX algorithm here included a new spectral flow cytometry study (Dataset 1) and three existing mass cytometry datasets (Dataset 2, Dataset 3, and Dataset 4). However, some of the more critical questions are as follows:

1. T-REX algorithm was a modular data analysis workflow including UMAP, KNN, and MEM. This is not a new algorithm but a combination.

2. Why choose this workflow, UMAP, KNN, and MEM are traditional classical algorithm, and not compare it with another algorithm? Refer to "A comparison framework and guideline of clustering methods for mass cytometry data".

3. What is the basis for setting the parameters, and what data can be used?

4. The references are not normative, for example, reference 2.

5. The figures are not clearly marked.

*Reviewer #2 (Recommendations for the authors):*

For the rhinovirus study, much of the data are shown through visualization on UMAP plots and the change in tet+ cells is graphed. Perhaps it would also be informative to graph the data as percent of CD4^+^ cells across all the participants for both day 0 and day 7 to summarize the results in a single graph and to note the actual frequency of these cells.

Perhaps the T-REX method could be tested with data sets from functional assays (such as flow cytometric intracellular cytokine staining) that are more commonly used to detect and characterize antigen-specific T cells. For such data, could the method perhaps compare the stimulation condition to the unstimulated condition rather than samples from different time points?

*Reviewer #3 (Recommendations for the authors):*

- Please include additional details on how the enrichment of tertramer positive cells in "hotspot" regions is determined.

- A formal mathematical definition of the two measures: Degree of change and Direction of change will be greatly helpful.

- A table summarizing the details of each dataset (number of cells, markers, individuals, cells per individual…) will be greatly helpful for the readers.

- The two shades of purple in Figure 3A are indistinguishable.

---

## [Author Response]

Reviewer #1 (Recommendations for the authors):The machine learning workflow Tracking Responders EXpanding (T-REX) was created to identify changes in both very rare and common cells in diverse human immune monitoring settings. T-REX analysis of paired blood samples provides an approach to rapidly identify and characterize mechanistically significant cells and to place emerging diseases into a systems immunology context. Data types used to challenge the T-REX algorithm here included a new spectral flow cytometry study (Dataset 1) and three existing mass cytometry datasets (Dataset 2, Dataset 3, and Dataset 4). However, some of the more critical questions are as follows:1. T-REX algorithm was a modular data analysis workflow including UMAP, KNN, and MEM. This is not a new algorithm but a combination.

We appreciate this point and have included in the new Figure 7 a comparison of the T-REX algorithm to each of the individual points to show how the combination significantly outperforms each alone and how different choices impact the run time or accuracy of the algorithm.

2. Why choose this workflow, UMAP, KNN, and MEM are traditional classical algorithm, and not compare it with another algorithm? Refer to "A comparison framework and guideline of clustering methods for mass cytometry data".

We appreciate this point and had not initially made comparisons as other algorithms were designed for different problems than T-REX. The new Figure 7 makes these types of direct comparisons and shows that the formulation of T-REX used here is optimal in the setting of the rhinovirus data (where we have tetramer positivity that can be used to judge accuracy).

3. What is the basis for setting the parameters, and what data can be used?

We have added a section to the Discussion that explores both of these points in detail.

4. The references are not normative, for example, reference 2

We appreciate the point about normative references and would be happy to include any the reviewers or editor recommend. We also edited the first sentence of the introduction to more specifically refer to the type of approach used here (systems immune monitoring), which is a subset of the approaches that could fit under the term “systems immunology”. The new sentence reads:

“Single-cell systems immune monitoring approaches offer new ways to compare how an individual patient’s cells respond to treatment or change during infection.”

The references were:

1. Greenplate, A.R., Johnson, D.B., Ferrell, P.B., Jr. & Irish, J.M. Systems immune monitoring in cancer therapy. Eur J Cancer 61, 77-84 (2016).

2. Davis, M.M., Tato, C.M. & Furman, D. Systems immunology: just getting started. Nat Immunol 18, 725-732 (2017).

To this we have added the following two references:

Schultze, J. Teaching 'big data' analysis to young immunologists. Nat Immunol 16, 902–905 (2015).

Chattopadhyay, P. K., Gierahn, T. M., Roederer, M., and Love, J. C. (2014). Single-cell technologies for monitoring immune systems. Nature immunology 15, 128-135.

Older references discussing systems immunology, such as Mohler et al., Proc of the IEEE 1980 "A Systems Approach to Immunology", focus on mechanistic modeling of immune cell interactions using math modeling (which is not the focus of the types of human systems immune monitoring studies here). Our goal here was to introduce people to the data-driven and discovery-oriented approaches like those where we envision T-REX being used, but we welcome suggestions on any normative references that would improve the manuscript.

5. The figures are not clearly marked.

We have aimed to clarify figure markings in response to reviewers.

Reviewer #2 (Recommendations for the authors):For the rhinovirus study, much of the data are shown through visualization on UMAP plots and the change in tet+ cells is graphed. Perhaps it would also be informative to graph the data as percent of CD4^+^ cells across all the participants for both day 0 and day 7 to summarize the results in a single graph and to note the actual frequency of these cells.

We appreciate this point that it would be useful to see the tetramer+ cells as a percentage of CD4+ T cells across participants and days. This information is now included in a new Supplementary File 2 and referenced in the Results section after all the rhinovirus study subjects are introduced.

Perhaps the T-REX method could be tested with data sets from functional assays (such as flow cytometric intracellular cytokine staining) that are more commonly used to detect and characterize antigen-specific T cells. For such data, could the method perhaps compare the stimulation condition to the unstimulated condition rather than samples from different time points?

We appreciate this point and have tested T-REX on many more datasets, including comparisons of unstimulated conditions to stimulation conditions. T-REX is well-suited for this type of data as well and this information has been included in the text. However, we believe including additional biological studies is beyond the scope of the present study.

Reviewer #3 (Recommendations for the authors):- Please include additional details on how the enrichment of tertramer positive cells in "hotspot" regions is determined.

We have added more details on how the cutoff for enrichment of tetramer positive cells in the hotspot regions is set in Results section, “T-REX identifies cells in phenotypically distinct regions of significant change.”

- A formal mathematical definition of the two measures: Degree of change and Direction of change will be greatly helpful.

We appreciate this point and have included more detailed explanations of both degree of change and direction of change in the Results.

- A table summarizing the details of each dataset (number of cells, markers, individuals, cells per individual…) will be greatly helpful for the readers.

We agree that this information is helpful for readers. Supplementary File 1 includes a summary of individuals per dataset as well as cells per individual. Supplementary File 4 includes the antibody panel used for the rhinovirus dataset.

- The two shades of purple in Figure 3A are indistinguishable.

We have addressed this concern in Figure 4 by changing one of the colors from purple to green.